



# Multi-decadal fluctuations in root zone storage capacity through vegetation adaptation to hydro-climatic variability has minor effects on the hydrological response in the Neckar basin, Germany.

Siyuan Wang[1], Markus Hrachowitz[1], Gerrit Schoups[1]

[1]Department of Water Management, Faculty of Civil Engineering and Geosciences, Delft University of Technology, Stevinweg 1, 2628CN Delft, Netherlands

*Correspondence to:* Siyuan Wang (S.Wang-9@tudelft.nl)

**Abstract.** Climatic variability can considerably affect the catchment-scale root zone storage capacity ($S_{umax}$) which is a critical factor regulating latent heat fluxes and thus the moisture exchange between land and atmosphere as well as the hydrological response and biogeochemical processes in terrestrial hydrological systems. However, direct quantification of changes in $S_{umax}$ over long time periods and the mechanistic drivers thereof at the catchment-scale are missing so far. As a consequence, it remains unclear how climatic variability, such as precipitation regime or canopy water demand, affects $S_{umax}$ and how fluctuations in $S_{umax}$ may influence the partitioning of water fluxes and therefore, also affect the hydrological response at the catchment-scale. The objectives of this study in the Upper Neckar river basin in Germany are therefore to provide a detailed analysis of multi-decadal changes in $S_{umax}$ that can be observed as a result of changing climatic conditions over a 70-year period and how this further affects hydrological dynamics. More specifically, we test the hypotheses that (1) $S_{umax}$ significantly changes over multiple decades reflecting vegetation adaptation to climate variability, (2) changes in $S_{umax}$ are a dominant control on the evaporative index $I_E = E_A/P$ and thus on the partitioning of water into drainage and evaporative fluxes as described by deviations $\Delta I_E$ from parametric Budyko curves over time, (3) changes in $S_{umax}$ also affect short term hydrological response dynamics and a time-dynamic implementation of $S_{umax}$ as parameter in a hydrological model can improve the performance of a hydrological model.

In this study, based on long-term daily hydrological records (1953-2022) and a stepwise approach over multiple consecutive 20-year periods, we found that variability in hydroclimatic conditions, with aridity index $I_A$ (i.e. $E_P/P$) ranging between ~ 0.9 and 1.1 over the study period was accompanied by deviations $\Delta I_E$ between -0.02 and 0.01 from the expected $I_E$ inferred from the long-term parametric Budyko curve. Similarly, fluctuations in $S_{umax}$, ranging between ~ 95 and 115 mm or 20%, were observed over the same time period. While uncorrelated with long-term mean precipitation and potential evaporation, it was shown that the magnitude of $S_{umax}$ is controlled by the ratio of winter or summer precipitation ($p < 0.05$). In other words, $S_{umax}$ in the study region does not depend on the overall wetness condition as for example expressed by $I_A$, but rather on how water supply by precipitation is distributed over the year. However, fluctuations in $S_{umax}$ were found to be uncorrelated with observed changes in $\Delta I_E$. Consequently, replacing a long-term average, time-invariant estimate of $S_{umax}$ with a time-variable, dynamically changing formulation of that parameter in a hydrological model did not result in an improved representation of



the long-term partitioning of water fluxes, as expressed by $I_E$ (and fluctuations $\Delta I_E$ thereof), nor in an improved representation of the shorter-term response dynamics.

Overall, this study provides quantitative mechanistic evidence that $S_{umax}$ significantly changes over multiple decades reflecting vegetation adaptation to climatic variability. However, this temporal evolution of $S_{umax}$ cannot explain long-term fluctuations in the partitioning of water (and thus latent heat) fluxes as expressed by deviations $\Delta I_E$ from the parametric Budyko curve over multiple time periods with different climatic conditions. Similarly, it does not have any significant effects on shorter term hydrological response characteristics of the upper Neckar catchment. This further suggests that accounting for temporal evolution of $S_{umax}$ with a time-variable formulation of that parameter in a hydrological model does not improve its ability to reproduce the hydrological response and may therefore be of minor importance to predict the effects of a changing climate on the hydrological response in the study region over the next decades to come.

## 1 Introduction

Vegetation is a key component of the terrestrial hydrological cycle as it shapes the hydrological functioning of catchments by regulating the long-term average partitioning of water into drainage and evaporative fluxes (i.e. latent heat), frequently expressed as runoff ratio $C_r = Q/P$ [-] and evaporative index $I_E = 1 – Q/P = E_A/P$ [-], respectively. More specifically, vegetation transpiration, that in spite of uncertainties (Coenders-Gerrits et al., 2014) globally constitutes the largest fraction of all evaporative fluxes (Jasechko, 2018), is systematically controlled by the interplay between canopy water demand and water supply from the subsurface (Donohue et al., 2007; Yang et al., 2016b; Jaramillo et al., 2018a; Mianabadi et al., 2019). To survive, vegetation needs continuous access to water stored in the subsurface and accessible to roots to satisfy its canopy water demand. As a consequence, the vegetation present at any moment, and in particular its active root system, reflects its successful adaptation to the prevalent climatic conditions in a region (Laio et al., 2001; Schenk and Jackson, 2002; Rodriguez-Iturbe et al., 2007; Donohue et al., 2007; Gentine et al., 2012; Liancourt et al., 2012). Irrespective of geometry, distribution or structure of root systems, the *maximum* vegetation-accessible water storage volume in the unsaturated root zone of the subsurface, hereafter referred to as root zone storage capacity $S_{umax}$ [mm], represents the hydrologically relevant information of root systems (Rodriguez-Iturbe et al., 2007; Nijzink et al., 2016a; Savenije and Hrachowitz, 2017; Gao et al., 2024).

As a central part of hydrological systems, $S_{umax}$ is also a critical parameter in hydrological and land-surface models. As such, it can, in principle, be estimated as a function of root depths and the subsurface pore volume between field capacity and permanent wilting point (Scrivner and Ruppert, 1970; Sivandran and Bras, 2012, 2013). However, these data are typically not available at sufficient levels of detail. Alternatively, catchment-scale $S_{umax}$ can be estimated by three broad approaches. Firstly, it can be obtained by calibration as parameter of a hydrological model (Nijzink et al., 2018; Bouaziz et al., 2020; Wang et al., 2023; Sriwongsitanon et al., 2023; Roberts et al., 2021; Bahremand and Hosseinalizadeh, 2022; Sadayappan et al., 2023; Tong et al., 2022). Secondly, based on optimality principles, there are some variables like transpiration, nitrogen uptake or carbon gain that can be maximized to quantify $S_{umax}$ (Guswa, 2008; McMurtrie et al., 2012; Sivandran and Bras, 2012; Yang, et al.,





2016b; Speich et al., 2018). Thirdly, $S_{umax}$ can be robustly estimated at the catchment scale directly from annual water deficits
based on observed hydro-climatic data, i.e. precipitation and transpiration (e.g., Donohue et al., 2012; Gentine et al., 2012;
Gao et al., 2014b; De Boer-Euser et al., 2016; Nijzink et al., 2016a; Dralle et al., 2021; McCormick et al., 2021; Hrachowitz
et al., 2021; Stocker et al., 2023; van Oorschot et al., 2021, 2023). For applications of hydrological and land-surface models
$S_{umax}$ (or equivalent parameters) has, except for very few exceptions (Wagener et al., 2003; Merz et al., 2011; Bouaziz et al.
2022; Tempel et al., 2024) been assumed constant over time. As a major knowledge gap, it remains so far unknown if $S_{umax}$
follows climatic variability and evolves over time, thereby reflecting vegetation adaptation to changing conditions.

In contrast, it is well understood that, due to the importance of vegetation for the hydrological functioning of terrestrial systems,
anthropogenic land use management practices, such as de- and afforestation (Brown et al., 2005; Brath et al., 2006; Fenicia et
al., 2009; Alila et al., 2009; Jaramillo et al., 2018a; Teuling et al., 2019; Stephens et al., 2021; Hoek van Dijke et al., 2022;
Ellison et al., 2024) or irrigation (e.g. AghaKouchak et al., 2015; Van Loon et al., 2016; Roodari et al., 2021) can induce major
shifts in the partitioning between the major components of the terrestrial water and energy cycles, and thus between $I_E$ and $C_r$.
Two detailed recent studies with well documented information on deforestation in several experimental catchments could
establish explicit mechanistic links between the reduction of $S_{umax}$ by > 50% following deforestation and decreases in $I_E$ (and
thus increases in $C_r$) from $\sim 0.4 - 0.5$ to $\sim 0.1 - 0.3$, depending on the catchment and the scale of deforestation (Nijzink et
al.,2016a; Hrachowitz et al., 2021).

Mapping the shifts to lower $I_E$ that followed these land conversions from forest to grass- and rangeland type vegetation as a
function of the aridity index $I_A = E_P/P$ in the Budyko framework (Schreiber, 1904; Ol'Dekop, 1911; Budyko, 1974) corresponds
well to the results of previous studies that suggest that, across the world, catchments dominated by grass exhibit consistently
lower $I_E$ at the same $I_A$ than forest environments (e.g. Zhang et al., 2001, 2004; Oudin et al., 2008). These differences in long-
term average $I_E$ are accounted for by parametric reformulations of the Budyko framework, such as the Tixeront-Fu equation
(Tixeront, 1964; Fu, 1981). The lumped parameters (here: $\omega$) of these expressions define long-term average catchment-specific
positions in the $I_A - I_E$ space. As such, the parameters are typically interpreted to encapsulate vegetation characteristics and all
other hydro-climatic and physiographic properties of individual catchments besides $I_A$ (e.g. Roderick and Farquhar, 2011;
Berghuijs and Woods, 2016). A frequent assumption is that with changes in climatic conditions, here represented by $I_A$,
individual catchments can be expected to move to the associated new positions $I_E$, following their specific trajectories defined
by $\omega$ (e.g. Zhou et al., 2015; Bouaziz et al., 2022). However, several studies have demonstrated that catchments in many
regions world-wide experience deviations $\Delta I_E$ from their expected new $I_E$ following a change in $I_A$ (e.g. Jaramillo and Destouni
2014; van der Velde et al., 2014; Jaramillo et al., 2018a; Reaver et al., 2022; Tempel et al., 2024).

From the above the following questions arise: (1) following the notion that vegetation, i.e. individual plants but also the species
composition of plant communities, continuously adapts to climatic conditions, does catchment-scale root zone storage capacity
$S_{umax}$ change over multi-decadal time scales? (2) do multi-decadal changes in the vegetation response, expressed by changes
in $S_{umax}$, explain deviations $\Delta I_E$ from expected $I_E$? (3) does a time-variable representation of $S_{umax}$ as parameter in a hydrological
model improve the models' ability to reproduce the hydrological response?



Building on previous studies, the objectives of this study in the Upper Neckar river basin in Germany are therefore to provide an analysis of multi-decadal changes in $S_{umax}$ as a result of changing climatic conditions over a 70-year period (1953 – 2022)

and how this further affects hydrological dynamics. More specifically, we test the hypotheses that (1) $S_{umax}$ significantly changes over multiple decades reflecting vegetation adaptation to climatic variability, (2) changes in $S_{umax}$ affect the long-term partitioning of drainage and evaporation and thus control deviations $\Delta I_E$ from the catchment-specific trajectory in the Budyko space and (3) a time-dynamic implementation of $S_{umax}$ improves the representation of streamflow in a hydrological model.

## 2 Study area

The Upper Neckar River basin in South-West Germany covers an area of ~4000 km$^2$, with the Black Forest on the western part and the Swabian Jura on the southeastern side. The river basin has a varying topography with the elevation ranging from 250 m at the outlet in the north to about 1019 m in the South (Fig. 1a; Table 1). Following the elevation gradient, the landscape is dominated by terrace-like elements and undulating hills (~50%) with wide valleys used as grass- and croplands in lower regions, in particular in the southeastern parts of the Upper Neckar Basin, and increasingly steep and narrow forested valleys

(~40%) towards the southern parts and the remaining area including flat grassland in valley bottoms (~10%) (Fig. 1c). Annual mean precipitation (P) over the whole river basin has a considerable spatial heterogeneity ranging from ~700 mm yr$^{-1}$ in the lower parts of the basin to ~1600 mm yr$^{-1}$ over the Black Forest with catchment average long-term mean precipitation (P) reaching ~880 mm yr$^{-1}$ (Fig. 1b, Table 2). The catchment is characterized by a temperate-humid climate, with warm, wet summers and cold, drier winters. Precipitation exhibits some seasonality with ~500 mm yr$^{-1}$ for summer months (from May to

October) and ~380 mm yr$^{-1}$ for winter months (from November to April), respectively (Fig 3). Although snow is in general not a major component of precipitation in the study region, snowmelt can have a significant influence during individual storm events. The long-term mean temperature is about 8.2 °C and potential evaporation ($E_P$) is around ~860 mm yr$^{-1}$ with an aridity index $I_A = E_P/P$ ~0.97 (Table 2).

## 3 Data sets

### 3.1 Data

Daily hydro-meteorological data were available for the period 01/01/1953 – 31/12/2022 (Fig. 2). Daily precipitation and daily mean air temperature were obtained from stations operated by the German Weather Service (DWD). Precipitation was recorded at 15 stations and temperature measurements were available at 8 stations (Fig. 1) in or close to the study basin. Daily potential evaporation $E_P$ (mm d$^{-1}$) was estimated using the Hargreaves equation. Daily mean discharge data for the period 01/01/1953

125    – 31/12/2022 at the outlet of the upper Neckar basin at Plochingen station were provided by the German Federal Institute of Hydrology (BfG). In addition, data of daily mean discharge for the same time period from three sub-catchments within the upper Neckar basin (Fig.1) at the gauges Rottweil (C1; 422 km$^2$), Plochingen at Fils river (C2; 706 km$^2$) and Horb (C3; 1111



km$^2$) were available from the Environmental Agency of the Baden-Württemberg region (LUBW).

Based on the CORINE Land Cover data set of the upper Neckar river basin during the period 01/01/1953 – 31/12/2022
(https://land.copernicus.eu/pan-european/corine-land-cover), there is only very minor change (< 2%) for all defined land cover
classes (Fig. 1c). The 90 m × 90 m digital elevation model of the study region (Fig. 1a) was obtained from the HDMA database
of the USGS (Verdin, 2017; https://doi.org/10.5066/F7S180ZP) and used to derive the local topographic indices including
height above nearest drainage (HAND) and slope.

### 3.2 Data pre-processing

For the subsequent experiment (section 4.2), the study basin was stratified into three zones P1 – P3 that are characterized by
distinct long-term precipitation pattern (hereafter: precipitation zones), following the approach described and implemented for
the Neckar basin by Wang et al. (2023). Briefly, Goovaerts (2000) and Lloyd (2005) showed that areal precipitation estimates
informed by elevation data were often more accurate than those based on precipitation gauge observations alone (e.g.
Hrachowitz and Weiler, 2011). Thus, to interpolate and to estimate areal precipitation across the basin we used Co-Kriging,
considering elevation, as a preliminary analysis suggested lower errors. Finally, the individual precipitation estimates for each
grid cell were used with K-means clustering to establish three clusters, representing the three precipitation zones P1 – P3 (see
Fig. 1b).

### 4 Methods

To test the hypotheses that the key vegetation parameter, i.e. the root zone storage capacity $S_{umax}$, evolves over multi-decadal
timescales in response to changing hydro-climatic conditions and controls the deviations from expected trajectories in the
Budyko space, thereby reflecting the need for time-variable implementations of $S_{umax}$ as parameter in a hydrological model,
the following stepwise approach is designed: (1) Estimate the *observed* deviations $\Delta I_E$ from the long-term average expected $I_E$
for four consecutive periods $t_1 – t_4$ in the study period (Table 2), (2) Estimate the root zone storage capacity over the entire
study period ($S_{umax,WBT}$) as well as for the four individual periods $t_1 – t_4$ ($S_{umax,WBt}$) based on *observed* water balance data, (3)
Estimate the root zone storage capacity over the entire study period ($S_{umax,calT}$) and the four individual periods $t_1 – t_4$ ($S_{umax,cal,t}$)
by *calibration* of a hydrological model over the respective time periods to evaluate whether the changes in calibrated $S_{umax,cal}$
reflect changes in $S_{umax,WB}$ directly estimated from water balance data from step (2), (4) Estimate the *modelled* deviations
($\Delta I_{E,mT,O'}$) from expected $I_E$ using both, a long-term average time-invariant $S_{umax,WBT}$ and individual $S_{umax,WBt}$ for the four periods
$t_1 – t_4$ as model parameters.

### 4.1 Estimation of the temporal trajectory in the Budyko framework

Mapping aridity $I_A = E_P/P$, where $E_P$ is potential evaporation [mm d$^{-1}$] and P is precipitation [mm d$^{-1}$], against the evaporative





index $I_E = E_A/P = 1 - Q/P$, where $E_A$ is actual evaporation [mm d$^{-1}$] and Q is stream flow [mm d$^{-1}$], the Budyko framework is an expression of the long-term average water balance for a catchment. It is based on the assumption of negligible storage

change over the averaging time period, i.e. dS/dt ~ 0. As demonstrated by Han et al. (2020), this assumption holds for averaging periods ≥ 10 years for a large majority of catchments worldwide. Note, that hereafter the term evaporation is used to refer to all combined evaporative fluxes, including interception and soil evaporation ($E_i$) as well as transpiration ($E_T$), following the terminology proposed by Savenije (2004) and Miralles et al. (2020).

The analysis in this paper is based on the parametric Tixeront-Fu formulation of the Budyko framework (Tixeront, 1964; Fu,

1981):

$$I_{E,T} = 1 + I_{A,T} - (1 + I_{A,T}{}^{\omega_T})^{1-\omega_T} \tag{1}$$

where $I_{E,T}$ is the observed evaporative index based over a chosen averaging period T, $I_{A,T}$ is the observed aridity index over the same period and $\omega_T$ is the associated catchment-specific parameter that represents all combined catchment properties other than $I_A$.

In a theoretical catchment that only experiences changes in $I_A$ and no changes in any other hydro-climatic and/or physical catchment characteristics, it can be assumed that $\omega_T$ remains constant over time so that $\omega_T = \omega_{t_i} = \omega_{t_{i+1}}$. This implies that following a disturbance $\Delta I_A$ in a subsequent time period $t_{i+1}$ the catchment stays on its specific curve defined by $\omega_T$, to a new $I_{Et_{i+1}}$. In such a case, $\omega_T$ can thus be used to predict future hydrological partitioning $I_E$. Based on this assumption, we here use the complete available hydro-climatic data record to estimate the long-term average $\omega_{OT}$ as reference over the entire 1953 –

2022 study period. The sub-division into the four time periods $t_1 – t_4$ as shown in Table 2, then allowed to estimate the expected $I_{E,t_i}$' in the individual periods $t_1 – t_4$: depending on the shift in the observed aridity index along the x-axis in $t_i$ ($\Delta I_{A,T,ti} = I_{A,t_i} - I_{A,T}$), a catchment will move along its parametric Budyko curve defined by $\omega_{OT}$ to a new expected position $I_{E,t_i}$' (Fig. 4).

Based on the available data we then estimate the individual observed $I_{E,t_i}$ together with the associated $\omega_{t_i}$ for each of the four time periods $t_1 – t_4$ (Fig. 5) For each of the four time periods $t_1 – t_4$ the deviation of $I_{E,t_i}$ from the catchment-specific expected

$I_{E,t_i}$', corresponding to a shift from $\omega_T$ to $\omega_{t_i} \neq \omega_T$ was then computed as $\Delta I_{E,t_i,t_i'} = I_{E,t_i} - I_{E,t_i'}$.

**4.2 Estimation of root zone storage capacity derived by water balance method S$_{umax,WB}$**

The root zone storage capacity is the *maximum* volume of water which can be held in soil pores of the unsaturated zone and which is accessible to root systems of vegetation for transpiration. Here the water balance method that is in detail described in previous papers (e.g. Gao et al., 2014b; Nijzink et al., 2016a; de Boer-Euser et al., 2016; Wang-Erlandsson et al., 2016; Bouaziz

et al., 2020; Hrachowitz et al., 2021) is used to determine S$_{umax,WB}$. Briefly, S$_{umax,WB}$ is estimated based on daily observations of precipitation P, potential evaporation $E_P$ and stream flow Q. As a first step, effective precipitation $P_e$ [mm d$^{-1}$] that enters the subsurface is computed by accounting for interception evaporation by:

$$P_e(t) = P(t) - E_i(t) - dS_i/dt \tag{2}$$

where $E_i$ (mm d$^{-1}$) is daily interception evaporation, $S_i$ (mm) is the interception storage. For each time step, $E_i$ is





determined by:

$$E_i(t) = \begin{cases} E_P(t) & if\ E_P dt < S_i \\ \dfrac{S_i}{dt} & if\ E_P dt \geq S_i \end{cases} \tag{3}$$

Then further to estimate the effective precipitation $P_e$ (mm d$^{-1}$) according to:

$$P_e(t) = \begin{cases} 0 & if\ S_i < S_{imax} \\ \dfrac{S_i - S_{imax}}{dt} & if\ S_i \geq S_{imax} \end{cases} \tag{4}$$

where $S_{imax}$ (mm) is the maximum interception storage, here assumed to be 2 mm (de Boer-Euser et al., 2016; Bouaziz et al., 2022).

Hereafter, the long-term mean transpiration $\overline{E_r}$ is estimated from the long-term water balance, with the assumption of no additional gains or losses:

$$\overline{E_r} = \overline{P_e} - \overline{Q_O} \tag{5}$$

where $\overline{P_e}$ (mm d$^{-1}$) is the long-term mean effective precipitation and $\overline{Q_O}$ (mm d$^{-1}$) is the long-term mean observed streamflow. Considering the seasonal fluctuation of energy input, the daily transpiration $E_r$ (mm d$^{-1}$) is estimated by subsequently scaling the daily potential evaporation minus the interception evaporation by the long-term mean $E_r$, according to (Bouaziz et al., 2022, Hrachowitz et al., 2021):

$$E_r(t) = \frac{\overline{E_r}}{(\overline{E_P - E_i})}(E_P - E_i) \tag{6}$$

From daily storage deficits $S_{rd,n}$ (t) (mm) during dry periods, estimated as the cumulative sum of daily effective precipitation $P_e$ (mm d$^{-1}$) minus transpiration $E_r$ (mm d$^{-1}$), the maximum storage deficit $S_{rd,n}$ of a specific year is then computed as follows:.

$$S_{rd,n}(t) = f(x) = \begin{cases} \int_{t_{0,w}}^{t_{0,d}}(P_e(t) - E_r(t))dt\,, & if\ S_{rd}(t) \leq 0 \\ 0, & if\ S_{rd}(t) > 0 \end{cases} \tag{7}$$

$$S_{rd,n} = \max\left(\left|S_{rd,n}(t)\right|\right) \tag{8}$$

Where t is the time step (d), $t_{0,w}$ is the day at the end of the wet period when the storage deficits are zero but $P_e(t) - E_r(t) < 0$, and $t_{0,d}$ is the day when storage deficits return to zero again after the begin of the next wet period when the water supply exceeds canopy water demand, i.e., $(P_e(t) - E_r(t)) > 0$. Any cumulative precipitation surplus is assumed to be drained from root zone and released from the system either directly as streamflow or via recharge of the groundwater.

The Gumbel extreme value distribution (Gumbel, 1941) was previously used for estimating the root zone storage capacity through the water balance approach by several other studies (Gao et al., 2014; Nijzink et al., 2016a; de Boer-Euser et al., 2016; Bouaziz et al., 2020, 2022; Hrachowitz et al., 2021). Based on fitting the Gumbel distribution to the maximum annual storage deficits for all n years during one of the four time periods t1 – t4, the root zone storage capacity $S_{umax,WB}$ can be derived from various return periods of the sequence of n maximum annual storage deficits $S_{rd}$. Previous studies suggested that vegetation develops root zone storage capacities large enough to survive in dry spells with return periods of ~ 20 – 40 years. Therefore,





we define $S_{umax,WB}$ here as the maximum storage deficit in a 40-year period so that $S_{umax,WB}=S_{rd,40yr}$.

Using the above water balance based method, we determine $S_{umax,WB}$ for the entire study period 1953 – 2022 ($S_{umax,WB,T}$) as well as individually for the four time periods $t_1 – t_4$ ($S_{umax,WB,t}$) to quantify potential fluctuations of root zone storage capacity reflecting the adaptation to changing climatic conditions.

## 4.3 Hydrological model

### 4.3.1 Model architecture

Loosely based on the flexible DYNAMITE modular modelling framework (e.g. Hrachowitz et al., 2014), we here used a semi-distributed, process-based model, that has previously been successfully implemented and tested for the Neckar study basin (Wang et al., 2023) and for many other contrasting environments world-wide (e.g. Prenner et al., 2018; Hulsman et al., 2021a,
c; Hanus et al., 2021; Bouaziz et al., 2022). Briefly, this hydrological model consists of three parallel hydrological response units (HRU), i.e. forest, grass/cropland and wetland, which are linked through a common storage component representing the groundwater system (Fig. 2). The classification into the three HRUs was based on the metric Height-above-nearest-drainage (HAND; Gharari et al., 2011) and land cover similar to previous studies (e.g. Gao et al., 2014a; Gharari et al., 2014; Nijzink
et al., 2016b; Bouaziz et al 2021). The model was further spatially discretized by a stratification into 100m-elevation bands for a more detailed representation of the snow storage ($S_{snow}$) and finally implemented in parallel, i.e. individually for each of the three precipitation zones P1 – P3 to balance to a certain degree spatial differences in precipitation with computational requirements. Rain ($P_{rain}$) and melt water ($M_{snow}$) from the different elevation zones were aggregated according to their associated spatial weights in each elevation zone as further input to the subsequent layers of the model in each HRU. The
outflows from each HRU in each precipitation zone as well as finally the outflows from each precipitation zone were likewise aggregated according to their respective spatial weights to represent the catchment aggregated outflows. While the three HRU's are characterized by distinct parameters that reflect their respective functioning, the parameters between the individual zones P1 – P3 were, in the spirit of model parsimony, kept the same in what is elsewhere referred to as a distributed moisture accounting approach (e.g. Ajami et al., 2004; Fenicia et al., 2008; Euser et al., 2015). Overall, the model consists of snow
($S_{snow}$), interception ($S_i$), unsaturated root zone ($S_u$), fast responding ($S_f$) and slow responding storage ($S_s$) components for each HRU and precipitation zone. The maximum storage volume in the unsaturated root zone component in each HRU is defined by the corresponding calibration parameters $S_{umax,F}$, $S_{umax,G}$ and $S_{umax,W}$, respectively. The catchment average $S_{umax,cal}$ is then inferred by aggregating these parameters according to their spatial weights. Water can be released from unsaturated root zones as combined soil evaporation and transpiration flux $E_t$ (mm d$^{-1}$) which is a frequently applied way to represent vegetation water
stress (e.g., Bouaziz et al., 2021, Gharari et al., 2013; Gao et al., 2014a). The equations of the model are provided as Table S1 in the Supplementary Material and more detailed descriptions of the model are provided by Wang et al. (2023) and other earlier implementations referred to above.





### 4.3.2 Model calibration

The model was run with a daily time step and has 18 calibration parameters. Briefly, the model parameters were calibrated by
255 using the Borg_MOEA algorithm (Borg Multi-objective evolutionary algorithm; Hadka and Reed, 2013) and based on uniform
prior distributions (Supplementary Material Table. S2). To best reflect different aspects of the hydrograph, including high flows,
low flows and the partitioning of precipitation into runoff and evaporation, the parameters are calibrated using a multi-criteria
approach that includes 7 objective functions as performance metrics $E_{Q,n}$ (Table 3). All solutions that fell on the 7-dimensional
front of pareto optimal solutions were kept as feasible. The 7 performance metrics were subsequently also combined into an
260 overall performance metric based on the Euclidian distance ($D_E$), where $D_E = 1$ indicates a perfect fit. To find a somewhat
balanced solution in absence of more detailed information all individual performance metrics were here equally weighted (e.g.,
Hrachowitz et al., 2021; Hulsman et al., 2021c; Wang et al., 2023):

$$D_E = 1 - \sqrt{\frac{\sum_{n=1}^{N}(1-E_{Q,n})^2}{N}} \tag{9}$$

where $N = 7$ is the number of performance metrics with respect to stream flow ($E_{Q,n}$). Note that the different units and thus
different magnitudes of residuals in the individual performance metrics introduce some subjectivity in finding the most
balanced overall solution according to $D_E$ (Eq. 9). However, a preliminary sensitivity analysis with varying weights for the
individual performance metrics in $D_E$ suggested limited influence on the overall results and is thus not further reported here.
In addition, the model was tested for its ability to represent spatial differences in the hydrological response by evaluating it
against streamflow observations in three sub-catchments (C1 – C3) of the upper Neckar catchment without further re-
270 calibration whereby each one of the sub-catchments largely represents the hydrological response from one of the precipitation
zones (Fig. 1).

The model is calibrated following two distinct calibration scenarios as indicated in Table 2. In the first scenario, the model and
thus also $S_{umax,F}$, $S_{umax,G}$ and $S_{umax,W}$ are calibrated over the full length of the 70-yr study period from 1953 – 2022. This reflects
the common assumption of a system that is stable over time. By extension, this also implies that the role of vegetation and thus
$S_{umax}$ does not change and that vegetation does not adapt to climatic variability. In the second scenario, individual calibration
to the four time periods $t_1 - t_4$ allowed to estimate fluctuations in the parameters $S_{umax,F}$, $S_{umax,G}$ and $S_{umax,W}$ between the time
periods as indicator of vegetation adaption to changing climatic conditions.

## 5 Results

### 5.1 Observed multi-decadal hydroclimatic variability

Based on the initial analysis of water balance data for four sub-time periods, significant differences were observed in the
variability of different hydroclimatic indicators over the 1953 – 2022 study period (Fig. 3). While periods $t_1$ and $t_4$ were
characterized by rather low mean annual precipitation of ~ 870 and 811 mm yr$^{-1}$, respectively, periods $t_2$ and $t_3$ were subject





to, on average, higher precipitation with ~ 911 mm yr$^{-1}$. While summer precipitation remained rather stable over the study period (Fig. 3f), the above was mostly caused by fluctuations in winter precipitation (Fig. 3k). In contrast, potential evaporation $E_P$ has gradually increased by 7% from 836 to 906 mm yr$^{-1}$ (Fig. 3b). Similarly reflecting increases in temperature (Fig. 3b), the annual snowpack and associated snowmelt have continuously decreased from around 98 mm yr$^{-1}$ to around 50 mm yr$^{-1}$ between $t_1$ and $t_4$ (Fig. 3c). A slight decrease of the number of days with precipitation from ~ 264 to 251 (Fig. 3d), on average, mostly due to changes in the summer months (Fig. 3n) was accompanied by some rather limited variability in precipitation intensities (Fig. 3e), mostly during winter (Fig. 3j). Overall, the comparatively humid periods $t_1 - t_3$ that were characterized by $I_A$ fluctuating between 0.93 – 0.97 were followed by a markedly more arid period $t_4$ with $I_A = 1.12$ (Table 2; Figure 6). In response to the multi-decadal variability in $I_A$, expressed as movement along the x-axis in the Budyko framework, the catchment experienced $I_E$ to vary between 0.56 and 0.59 (Table 2; Figure 6). However, this observed variability was somewhat lower than the variability $I_{E,\omega_T} = 0.55 – 0.61$ that would have been expected based on $\omega_T$. This illustrates that the hydrological response did not consistently follow its long-term trajectory defined by $\omega_T$. Instead, deviations $\Delta I_{E,t_i}$ from the expected positions, and thus values of $\omega_{t_i}$ that are different to $\omega_T$, were observed for the individual periods. More specifically, the deviations gradually decreased from $\Delta I_{E,t_1} = 0.01$ in $t_1$ to $\Delta I_{E,t_4} = - 0.02$ in $t_4$ (Fig. 6). This systematic shift towards lower (more negative) $\Delta I_{E,t_i}$ and thus also lower $\omega_{t_i}$ indicates that at the same $I_A$ a smaller fraction of precipitation is released as evaporation, i.e. $I_E$, now than at the start of the 70 year study period. Although the magnitude of deviations remains with $\Delta I_{E,t_i} \leq \pm 0.02$ rather minor, in particular their systematic shift into one direction implies that changes in the system other than $I_A$ have a visible effect on the hydrological response pattern.

## 5.2 Root zone storage capacity $S_{umax,WB}$ estimated from water balance data

As the baseline of our study, the annual maximum storage deficits fluctuate between 97 mm in 2022 and 16 mm in 1970 (Fig. 7a). Assuming an adaptation to dry spells with 40-yr return periods the root zone storage capacity over the entire 1953 – 2022 study period (Scenario 1) was estimated to $S_{umax,WB,T} = 105$ mm (Table 2; Fig. 7b). In the next step, the storage deficits and the associated root zone storage capacity for each period $t_1 - t_4$ was estimated (Scenario 2). $S_{umax,WB,t_1}$ and $S_{umax,WB,t_3}$ for periods $t_1$ and $t_3$, respectively, are estimated at the same value of 95 mm. In contrast, and somewhat counterintuitively, the highest value over the study period is found in the wettest period ($t_2$) and reaches $S_{umax,WB,t_2} = 115$ mm, while the driest period ($t_4$) is characterized by $S_{umax,WB,t_4} = 100$ mm (Table 2; Fig. 7c-j). These pattern suggest that $S_{umax,WB}$ did vary by ~20 mm, equivalent to ~20% throughout the 1953 -2022 period. In contrast to $\Delta I_{E,t_i}$ that was characterized by a systematic shift towards more negative deviations over time, no evidence was found for a systematic, one-directional shift in $S_{umax,WB}$. Instead, $S_{umax,WB}$ evolved following a somewhat cyclic pattern.





### 5.3 Root zone storage capacity $S_{umax,cal}$ estimated as calibration parameter

#### 5.3.1 Model calibration for 1953 – 2022 (Scenario 1)

The model parameter sets obtained as feasible after calibration over the entire 1953-2022 study period in Scenario 1 reproduce the main features of the hydrological response (Fig. 2d). More specifically, the modelled hydrographs in particular describe well the timing of high flows, albeit somewhat underestimating flow peaks for the best-performing model in terms of the $D_E$ (Eq. 9). The low flows and the shapes of recessions are in general well captured ($NSE_{logQ} = 0.67$). Crucially, the model also reproduces well the other observed stream flow signatures such as the flow duration curves ($NSE_{logFDC} = 0.96$), the autocorrelation function ($NSE_{AC} = 0.99$) as well as the long-term and seasonal runoff coefficients ($NSE_{Cr} = 0.90$, $RE_{Cr,summer} = 0.83$ and $RE_{Cr,winter} = 0.91$). The latter further implies that the modelled evaporative fluxes $E_A$ and thus $I_{E,\omega_T}$ are, on average, close to the observed ones, which can be seen in Figure 6. The model, calibrated on the overall response of the Upper Neckar basin, also exhibited considerable skill to represent spatial differences in the hydrological response by reproducing observed stream flow in the three sub-catchments (C1 – C3) similarly well (Fig. 8) without any further re-calibration. The overall model skill to mimic the hydrological response corresponds well to a similar implementation of the model in the greater study region by Wang et al. (2023). The detailed list of performance metrics is provided in Table S3 in the Supplementary Material.

The model calibration resulted in pronounced differences in the root zone storage capacity parameters for three individual landscape classes. While for forest dominated land it was estimated at $S_{umax,F} = 158$mm for the best performing model (5th/95th percentile of all feasible solutions: 138–168mm), it reached $S_{umax,G} = 95$mm (5th/95th: 71–123mm) for grass/cropland and $S_{umax,W} = 61$mm for wetland (5th/95th: 49–68mm), which reflects differences in vegetation type and position in the landscape (cf. Fan et al., 2017). Remarkably, the catchment root zone storage capacity, estimated by aggregating the individual values according to their areal fractions, came with $S_{umax,cal} = 116$mm (5th/95th : 99–130 mm, Fig. 9a) very close to the estimate $S_{umax,WB} = 105$ mm that is directly derived from water balance method without any calibration, as described in section 5.2.

#### 5.3.2 Model calibration for individual periods $t_1 – t_4$ (Scenario 2)

The model parameter sets obtained from the individual calibration for each period $t_1$ - $t_4$ reproduce the hydrographs of the corresponding periods as well or slightly better than when using the long-term average parameters from scenario 1 (see detailed performance metrics in Table S3 in the Supplementary Material). In particular, the runoff coefficients could with $NSE_{Cr}$ ~0.86 – 0.91, $RE_{Cr,summer}$ ~ 0.84 – 0.90 and $RE_{Cr,winter}$ ~ 0.88 – 0.92 be rather well mimicked. Similarly, the daily dynamics with $NSE_{logQ}$ ~ 0.63 – 0.72 for the best- performing model of each period. and most other hydrological signatures, could be reproduced marginally better.

The individual calibration over each period $t_1 – t_4$ resulted in associated differences in the catchment-scale root-zone storage capacity of each period. Based on the best-performing models, the calibrated values varied between low values for $t_1$ and $t_2$, with $S_{umax,cal,t_1} = 98$ mm and $S_{umax,cal,t_3} = 99$ mm and higher values for the two other periods with $S_{umax,cal,t2} = 122$ mm and $S_{umax,cal,t_4} = 107$ mm (Table 2; Figure 9a). The magnitudes of $S_{umax,cal,t_i}$ obtained by calibration in the individual time periods $t_1$



– $t_4$ are with a difference of 5 mm (~ 5%), on average, very close to $S_{umax,WB,t_i}$ estimated on basis of the water balance for the

same periods. Perhaps even more notably the temporal evolution of $S_{umax,cal,t_i}$ and $S_{umax,WB,t_i}$ follows the same sequence over time ($R^2 = 0.95$, $p = 0.05$; Figure 9b).

### 5.4 Effect of $S_{umax}$ on temporal fluctuation in the trajectories of the Budyko curve

The deviations $\Delta I_{E,O,O'}$ between expected evaporative index $I_{E,O'}$ and observed evaporative index $I_{E,O}$ (section 4.1.2) for all periods $t_1$ – $t_4$ become gradually more negative from $t_1$ ($\Delta I_{E,O,O'} = 0.013$) to $t_4$ ($\Delta I_{E,O,O'} = -0.020$), which is consistent with

decreases of $\omega_{t_i}$ and downward shifts of the associated parametric Budyko curves over time as described in section 5.1. These systematic reductions of $\Delta I_{E,O,O'}$ over the 70-year study period are not reflected in the fluctuations of root zone storage capacities, irrespective of how they were estimated, i.e. $S_{umax,WB,t_i}$ ($p = 0.85$) or $S_{umax,cal,t_i}$ ($p = 0.96$), as illustrated by Figure 10. The above is further corroborated by comparing the modelled $I_E$ from Scenarios 1 and 2 for each period $t_1$ – $t_4$. More specifically, in Figure 11(c) it can be seen that Scenario 1, based on a long-term average, time-invariant $S_{umax,WB,T}$ obtained over the entire

1953 – 2022 period, generates deviations $\Delta I_{E,mT,O'}$ from the expected long-term average $I_{E,O'}$ for each period $t_1$ – $t_4$. In this case, the modelled $I_E$ does not follow the expected $I_{E,O'}$. However, it also does not follow the sequence of increasingly negative deviations from 0.013 in $t_1$ to – 0.020 in $t_4$ as observed in reality ($\Delta I_{E,O,O'}$). Instead, $\Delta I_{E,mT,O'}$ remains negative for all time periods and fluctuates between $\Delta I_{E,mT,O'}$ = – 0.005 and – 0.029 (white boxplots in Fig. 11c). Replacing the time-invariant $S_{umax,WB,T}$ by individual $S_{umax,WB,t_i}$ for each period $t_1$ – $t_4$ in Scenario 2, accounts for the different effects of vegetation in the

individual periods. If $S_{umax}$ controlled the observed deviations from expected $I_{E,O'}$, Scenario 2 would generate estimates of $\Delta I_{E,mt_i,O'}$ that are closer to the observed ones than those of Scenario 1. However, no evidence was found for that: the deviations $\Delta I_{Emt,O'}$ obtained by Scenario 2 with time-variable $S_{umax}$ for each period $t_1$ – $t_4$ are largely indistinguishable (orange boxplots in Fig.11c) from those generated by Scenario 1 with time-invariant $S_{umax}$. As a consequence, the evaporative index $I_E$ modelled with time-variable $S_{umax,WB,t_i}$ is not found to be closer to the observed $I_E$ for Scenario 2 than for Scenario 1. On the contrary,

the deviations $\Delta I_{Em,O}$ from the observed $I_E$ obtained from the time-invariant Scenario 1 are in most time periods, albeit only slightly, less pronounced (Fig. 11d).

### 5.5 Effect of $S_{umax}$ on stream flow

Corresponding to the above findings, there is no significant difference in modelled average streamflow between Scenario 1,

using long-term average $S_{umax}$, and Scenario 2, using individual $S_{umax}$ values for each time period (Fig. 12d). While the model for both scenarios consistently and similarly underestimates high flows ($Q^{5th}$, Fig.12a) by ~10%, it overestimates median flow by ~15% with both time-invariant and time-variable $S_{umax}$, for all time periods (Fig.12b). Interestingly, the low flows are over-predicted by ~ 10 – 20% in the first two periods, while they are under-predicted by up to ~20% in the later periods in both Scenarios (Fig.12c). In addition, it was found that using time variable $S_{umax}$ in Scenario 2 did also not have any discernible





effect on seasonal flow pattern (not shown). The fact that both scenarios generate similar estimates over different flow percentiles and, in particular, that the time-variable Scenario 2 reflects the same systematic shift in the ability of the model to reproduce low flows as Scenario 1, suggests, together with the very minor effects of time-variable $S_{umax}$ in Scenario 2 on the model performance metrics, that the adaptation of $S_{umax}$ to changing climatic conditions does not significantly affect the average hydrological response pattern in the Neckar basin.

## 6 Discussion

### 6.1 Multi-decadal changes in root zone storage capacity $S_{umax}$

This study is the first to explicitly quantify how root zone storage capacity $S_{umax}$ changes with changing climatic conditions over time. The values of root zone storage capacity, estimated from both, water balance data and as model calibration parameter, show indeed significant and corresponding fluctuations over multiple decades, varying by up to ±20%. The overall estimated
magnitudes fall with $S_{umax} \sim 95 - 115$ mm well within the range of long-term average values reported previously for the greater region (e.g. Bouaziz et al., 2021; Hrachowitz et al, 2021; Tempel et al., 2024) and other temperate, humid environments (e.g. Kleidon, 2004; Gao et al., 2014b; de Boer et al., 2016; Wang-Erlandsson et al., 2016; Stocker et al., 2023; van Oorschot et al., 2023).

The values of $S_{umax}$ obtained from both methods are very similar and within an error margin of merely ~5%. In addition, they
both follow a comparable change over time. Together, this lends support to the underlying assumption that this temporal evolution of $S_{umax}$ may indeed be a fingerprint of vegetation adaptation to changing climatic conditions. More specifically, as $S_{umax,WB}$ is explicitly based on the estimates of transpiration $E_r$ (Eq. 9), it could be plausibly argued that during specific years merely more water is *used* for $E_r$ but that the size of the water storage volume accessible for roots may not necessarily change. In that case, changes in $S_{umax,WB}$ would not reflect actual changes of the active root system but only in how much water was
*used* by them. In contrast, $S_{umax,cal}$ inferred as calibration parameter of a hydrological model does not only regulate transpiration, but, critically, *also* the generation of streamflow. If therefore the active root system did in reality not change and fluctuations in $S_{umax,WB}$ were a mere artifact of changes in water uptake from a fixed-size volume instead of an actual change in of maximum vegetation-accessible subsurface water volumes, fluctuations in $S_{umax,cal}$ would not mirror those of $S_{umax,WB}$ and the use of $S_{umax,WB}$ in the hydrological model would, due to the non-linear character of the flow generation function in the model (Eq. 20
S1 in the Supplementary material), lead to misrepresentations of streamflow dynamics. Yet here, no deteriorations of the model performance with changing $S_{umax}$ were found. Even more, the fact that $S_{umax,WB}$ and $S_{umax,cal}$ are characterized by very similar magnitudes and fluctuations does add further evidence that their evolution over time is a manifestation of vegetation adapting its active root system to changing climatic conditions.

Several previous studies in similar environments found that the root zone storage capacity $S_{umax}$ can decrease by 50% or more
after deforestation and that these changes do not only cause reductions in $I_E$ by -0.2 or more, which reflect changes in ω and





thus changes of the overall functioning of the system, but also influence hydrological dynamics at short time scales, such as the magnitudes of flow peaks (Nijzink et al., 2016a; Hrachowitz et al.,2021). In contrast to the above studies, the ±20% fluctuations of $S_{umax}$ here did not lead to similarly marked shifts in $I_E$ or $\omega$. This is further corroborated by an analysis of different variables as potential controls on $S_{umax}$ and $\Delta I_E$ as shown in Figure 13. Fluctuations of $S_{umax}$ can to a large part be

attributed to the variability in the ratio of winter precipitation over summer precipitation (Fig. 13s) as simplified metric for precipitation seasonality. This comes as no surprise, as the computation of $S_{umax,WB}$ is explicitly based on the seasonal water deficit (Eq.7). It merely visualizes that the more precipitation falls in summer, in a time when evaporative demand is highest, the lower $S_{umax}$ needs to be to provide vegetation sufficient and continuous access to water for continuous vegetation transpiration. All other tested variables do not exert any major influence on $S_{umax}$ in the study region. Conversely, it was found

that the deviations $\Delta I_E$ are largely independent of the seasonality of precipitation (Fig. 13g). Instead, increases in summer $E_P$ are correlated with decreases in $\Delta I_E$ (Fig. 13h) and thus with a reduction of $E_T$. The observed systematic shift towards more negative $\Delta I_E$ which indicates proportionally less evaporation thus coincides with the gradually increasing summer $E_P$ over time. This points towards different controls on $\Delta I_E$ than on $S_{umax}$ and the potential role of increased vegetation water stress in summer as main driver of $\Delta I_E$. Thus, while there is compelling evidence for fluctuations in $S_{umax}$, the above illustrates that these changes

cannot explain the observed deviations from the expected long-term Budyko trajectory in the study region.

It is also important to note that the temporal fluctuations of both $S_{umax}$ and $\Delta I_E$ can be subject to uncertainties. In spite of the findings reported by Han et al. (2020), that for most river catchments world-wide dS/dt ∼ 0 holds over averaging periods similar to the ones used here ($t_1 – t_4$), this assumption may not completely hold in the study region. In relation with that, we also did not consider potential effects of unobserved groundwater import or export on the long-term water balance (Bouaziz

et al., 2018).

As only < 2% of the study area experienced documented land use change over the 1953 – 2022 period and no major reservoirs are present upstream of the study basin outlet, we here interpret fluctuations in $S_{umax}$ as a reflection of adaptation of root-systems to changing hydroclimatic conditions. However, some of the fluctuations may be the consequence of land management practices not quantified by available gridded land cover products such as CORINE, including forest thinning (cf. Hrachowitz

et al., 2021) or rejuvenation (cf. Teuling and Hoek van Dijke, 2020). In addition, although we here attribute changes in $S_{umax}$ mainly to changes in root systems, these may be complemented by additional effects of changes in vegetation water use due to feedbacks with increases in atmospheric $CO_2$ (e.g. Berghuijs et al., 2017; Jaramillo et al., 2018a).

## 6.2 Effect of changing $S_{umax}$ on the representation of stream flow in a model

Reflecting its lack of explanatory power for the changes in $\Delta I_E$, our results correspondingly indicate that signatures of both
annual flow, such as the average $Q^{5th}$, $Q^{50th}$ or $Q^{95th}$ but also of seasonal flow are not better reproduced by the hydrological model when replacing a time-invariant, long-term average $S_{umax}$ by a temporally dynamic $S_{umax}$. Overall, these results are in contrast to previous studies that quantified the effect of a time variable $S_{umax}$ parameter following deforestation. For example, Nijzink et al. (2016a) reported that adjusting parameter $S_{umax}$ to a lower value does improve a model's ability to reproduce





streamflow after deforestation. These findings were strongly supported by Hrachowitz et al. (2021), who found that post-
deforestation model recalibration resulted in lower $S_{umax}$ and a significantly better performance compared to using parameters
from pre-deforestation calibration. However, our results are also different to those reported by Duethmann et al. (2020), who
found that accounting for vegetation dynamics in a model in form of changing surface resistances to vegetation improved the
long-term performance of the model. Similarly, Bouaziz et al. (2022), who estimated future $S_{umax}$ based on projected future
hydro-climatic conditions. In a somewhat more humid environment, they found that an estimated ~25% future increase of
$S_{umax}$ from ~ 170 mm to 226 mm may lead to reductions in mean and maximum annual Q of ~ 5%. More pronounced effects
were reported at the intra-annual time scale, with reductions of autumn and winter Q by up to ~15%. This was accompanied
by up ~15% increases in summer evaporation and 10% decreases in winter groundwater levels. Irrespective of the additional
uncertainties in their study introduced by future projections, the much less pronounced effects we found in our analysis are
most likely a consequence of the lower absolute magnitude of $S_{umax}$ that remains below 115 mm in the study region. These
lower $S_{umax}$ values reflect lower storage requirements in summer, due to a precipitation pattern in the Neckar basin that is more
evenly spread throughout the year. In other words, the fact that here ~55 – 60 % of the annual precipitation falls in summer
(Fig. 3f, k) when it is needed most by vegetation due to high $E_P$, removes the need for larger $S_{umax}$ as water storage buffer to
allow vegetation to survive. However, the lower the magnitude of $S_{umax}$, the more frequently storage deficits can be overcome
by even rather small rainstorms and the less water is (or needs to be) stored. Thus even if the relative changes are similar
between Bouaziz et al. (2022) and our study, abundant summer precipitation causes absolute $S_{umax}$ fluctuations of less than
±20 mm over time in the Neckar. This in turn limits the influence of the changes on the hydrological response, which has wider
implications on the use of models in the Neckar basin and potentially in other temperate regions with similar hydro-climatic
characteristics. More specifically, it has been argued that a changing climate will affect the properties of terrestrial hydrological
systems (e.g. Stevens et al., 2020). As these properties are represented by typically time-invariant parameters in hydrological
or land surface models, accounting for changing system properties with time-variable formulations of parameters may facilitate
more reliable predictions. For many model parameters such a time-variable formulation to estimate their future values is not
trivial due to frequently insufficient data and a general lack of mechanistic understanding of the underlying processes. The
estimation of $S_{umax}$ and its temporal evolution based on observed historical or projected future water balance data opens an
opportunity to estimate its magnitude under future conditions for use in models. However, in contrast to the findings in other
regions (e.g. Merz et al., 2011) and as discussed above, adapting $S_{umax}$ to changing conditions in the Neckar basin does not
lead to improved modelled representations of the hydrological response. It is therefore plausible to assume that the use of a
time-variable parameter $S_{umax}$ does not substantially improve future predictions and is thus not necessarily required for at least
the next few decades to come and that the use of a long-term average $S_{umax}$, obtained either by calibration or based on the water
balance is sufficient in the Neckar basin and in hydro-climatically similar regions.



## 7 Conclusions

The catchment-scale root zone storage capacity ($S_{umax}$) is a critical factor affecting the moisture exchange between land and atmosphere as well as the hydrological response in terrestrial hydrological systems. However, as a major knowledge gap, it is unclear if $S_{umax}$ at the catchment-scale evolves over time, reflecting vegetation adaptation to changing climatic conditions. As a consequence, it also remains unclear how potential changes in $S_{umax}$ may affect the partitioning of water fluxes and as a consequence, the catchment-scale hydrological response. In this study, for the upper Neckar catchment, based on long-term daily hydrological data (1953 – 2022), we quantify and analyze how $S_{umax}$ dynamically evolves over multiple decades reflecting vegetation adaptation to climate variability and the effects on the hydrological system.

The main findings of our analysis are the following:

(1) $S_{umax}$ has fluctuated by $\pm$ 20 % between 95 and 115 mm, in response to climatic variability over the 70-year study period.

(2) Estimates of $S_{umax}$ obtained from both methods, i.e. based on water balance data and as model calibration parameter, respectively, were with differences of ~5% highly consistent with each other and correlated in time ($R^2$ = 0.95, p = 0.05). Findings (1) and (2) support the hypothesis that $S_{umax}$, even in temperate, humid climates such as in the Neckar basin, significantly changes over multiple decades, reflecting vegetation adaptation to climatic variability.

(3) The estimated fluctuations of $S_{umax}$ were inconsistent with the temporal sequence of observed deviations $\Delta I_E \sim \pm$ 0.02 from the expected $I_E$ over the study period ($R^2$ = 0.02, p = 0.85).

(4) As a consequence, replacing a long-term average, time-invariant parameter $S_{umax}$ in a hydrological model with a time variable formulation of $S_{umax}$ does not lead to a better representation of the observed $\Delta I_E$. Based on (3) and (4), the hypothesis that $S_{umax}$ affects the long-term partitioning of drainage and evaporation and thus controls deviations $\Delta I_E$ from the catchment-specific trajectory in the Budyko space therefore needs to be rejected for the Neckar basin.

(5) Replacing time-invariant $S_{umax}$ with a time-variable $S_{umax}$ in the hydrological model leads to only very minor improvements of the model to reproduce streamflow dynamics. The hypothesis that a time-dynamic implementation of $S_{umax}$ improves the representation of streamflow in the hydrological therefore also needs to be rejected for the Neckar basin.

Overall, our study is the first to systematically document the temporal evolution of $S_{umax}$, and although limited to the Neckar basin, it provides clear quantitative evidence that $S_{umax}$ can significantly change over multiple decades reflecting vegetation adaptation to climate variability. However, these changes do not cause deviations from the long-term average Budyko curve under changing climatic conditions. This implies that the temporal evolution of $S_{umax}$ does not control variation in the partitioning of water fluxes and has no significant effects on fundamental hydrological response characteristics of the Upper Neckar basin. As the use of time-variable $S_{umax}$ over the 70-year study period does not improve performance of the hydrological model, it can plausibly be assumed that in the study region the use of time-invariant $S_{umax}$ as model parameter will be sufficient for meaningful predictions over at least the next few decades.

*Code availability.* The model codes underlying this paper will be available online in the 4TU data repository (DOI:





10.4121/b75c9108-c5b8-4266-9b82-1ad08c76adcc). The equations used in the model are described in supplement.

*Data availability.* The meteorological and hydrological data used in this study can be obtained from German Weather Service (DWD) and the German Federal Institute of Hydrology (BfG).

*Author contributions.* SW, MH and GS designed the study, SW executed the experiments, all authors contributed to general idea, the discussion and writing of the manuscript.

*Competing interests.* At least one of the (co-) author is a member of the editorial board of Hydrology and Earth System Science.

*Acknowledgements.* We gratefully acknowledge financial support from China Scholarship Council (CSC).

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





**Table1.** Characteristics of the Neckar catchment in Germany

| Characteristics | |
|---|---|
| latitude (N) | 48°02′00″-48°46′59″ |
| longitude (E) | 8°18′45″-9°56′33″ |
| Area (km$^2$) | 3968 |
| Average annual precipitation (mm yr$^{-1}$) | 880 |
| Average annual temperature (°C) | 8.39 |
| Elevation range (m) | 250-1019 |
| Mean elevation (m) | 554 |
| Slope range (°) | 0-53 |
| Mean slope (°) | 5.80 |
| Forest dominated land (%) | 39.6 |
| Grass dominated land (%) | 49.6 |
| Wetland (%) | 10.8 |

**Table2.** Mean annual precipitation P, potential evaporation $E_P$, temperature $T_m$, aridity index $I_A$, evaporative index $I_E$, parameter ω for parametric Budyko framework, root zone storage capacity $S_{umax,WB}$ and $S_{umax,cal}$ based on respectively water balance data and hydrological model calibration for scenario 1 (entire time period T: 1953-2022) and scenario 2 (four sub- periods t1:1953-1972, t2:1973-1992, t3:1993-2012, and t4:2013-2022).

| | Scenario 1 | Scenario 2 | | | |
|---|---|---|---|---|---|
| | T (1953-2022) | t1 (1953-1972) | t2 (1973-1992) | t3 (1993-2012) | t4 (2013-2022) |
| P (mm yr$^{-1}$) | 876 | 870 | 907 | 915 | 811 |
| $E_P$ (mm yr$^{-1}$) | 867 | 836 | 840 | 884 | 906 |
| $T_M$ (°C) | 8.4 | 7.4 | 7.9 | 8.7 | 9.5 |
| $I_A$(-) | 0.97 | 0.96 | 0.93 | 0.97 | 1.12 |
| $I_E$ (-) | 0.57 | 0.58 | 0.56 | 0.56 | 0.59 |
| ω (-) | 1.95 | 2.01 | 1.98 | 1.93 | 1.89 |
| $S_{umax,WB}$ (mm) | 105 | 95 | 115 | 95 | 100 |
| $S_{umax,cal}$ (mm) | 116 | 98 | 123 | 99 | 107 |

Table 3. Signatures of flow and the associated performance metrics used for model calibration and evaluation. The performance metrics include the Nash–Sutcliffe efficiency (NSE) and the relative error (RE).

| Signature | Symbol | Performance metric |
|---|---|---|
| Time series of stream flow | Q | $NSE_Q$ |
| Time series of log(Q) | log(Q) | $NSE_{log(Q)}$ |
| Flow duration curve of log(Q) | $FDC_{log(Q)}$ | $NSE_{FDClog(Q)}$ |
| Seasonal runoff coefficient | $C_r$ | $NSE_{Cr}$ |
| Autocorrelation function of flow (AC) | AC | $NSE_{AC}$ |
| Runoff coefficient in summer | $C_{r,summer}$ | $RE_{Cr,summer}$ |
| Runoff coefficient in winter | $C_{r,winter}$ | $RE_{Cr,winter}$ |

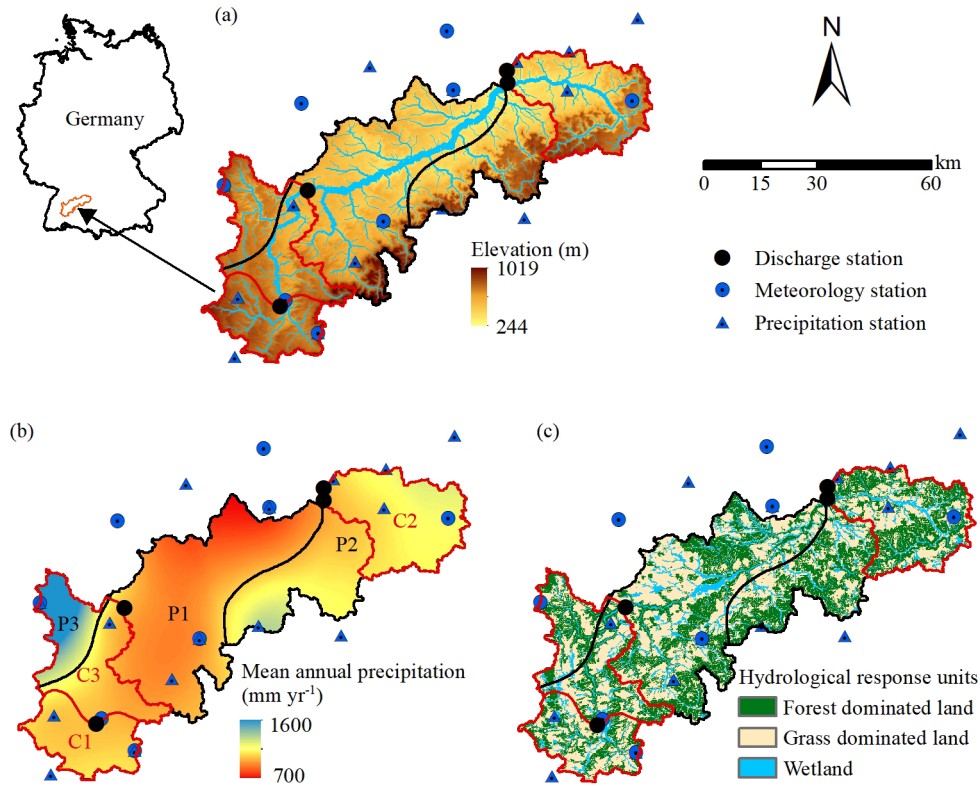

**Figure 1.** (a) Elevation of the Neckar catchment with discharge and hydro-meteorological stations as well as the water sampling locations used in this study, (b) the spatial distribution of long-term mean annual precipitation in the upper Neckar catchment and the stratification into three distinct precipitation zones P1 – P3 (black outline), and the red outlines indicate three sub-catchments (C1:Rottweil, C2: Plochingen at Files river, and C3: Horb) within the upper Neckar basin, (c) hydrological response units classified according to their land-cover and topographic characteristics.



**Figure 2.** (a) Time series of observed monthly precipitation P; (b) daily cumulative evaporative fluxes for entire time period (1953-2022), where the dark brown line indicates potential evaporation $E_P$ and the yellow lines and the light orange shaded areas show the actual evaporation $E_A$ modelled using the best fit parameter sets and the associated $5^{th}/95^{th}$ percentiles of all feasible solutions calibrated based on entire time period; (c) monthly maximum values of snow water equivalent (SWE) for 1953-2022 time period where green line indicates the most balanced solution and light green shade indicates the $5^{th}/95^{th}$ inter-quantile range obtained from all pareto optimal solutions calibrated based on entire time period; (d) observed (blue line) and modelled daily streamflow Q; red line indicates the most balanced solution and the shaded area indicates the $5^{th}/95^{th}$ percentile of all feasible solutions calibrated based on entire time period, respectively; the different green background shades from lighter to darker indicate sub-time periods from $t_1$ to $t_4$; (e) and (f) zoom-in to the observed and modelled streamflow for the selected wet year(light gray shade, 01/01/1988 – 31/12/1988) and dry year (gray shade, 01/01/2003 – 31/12/2003) respectively.

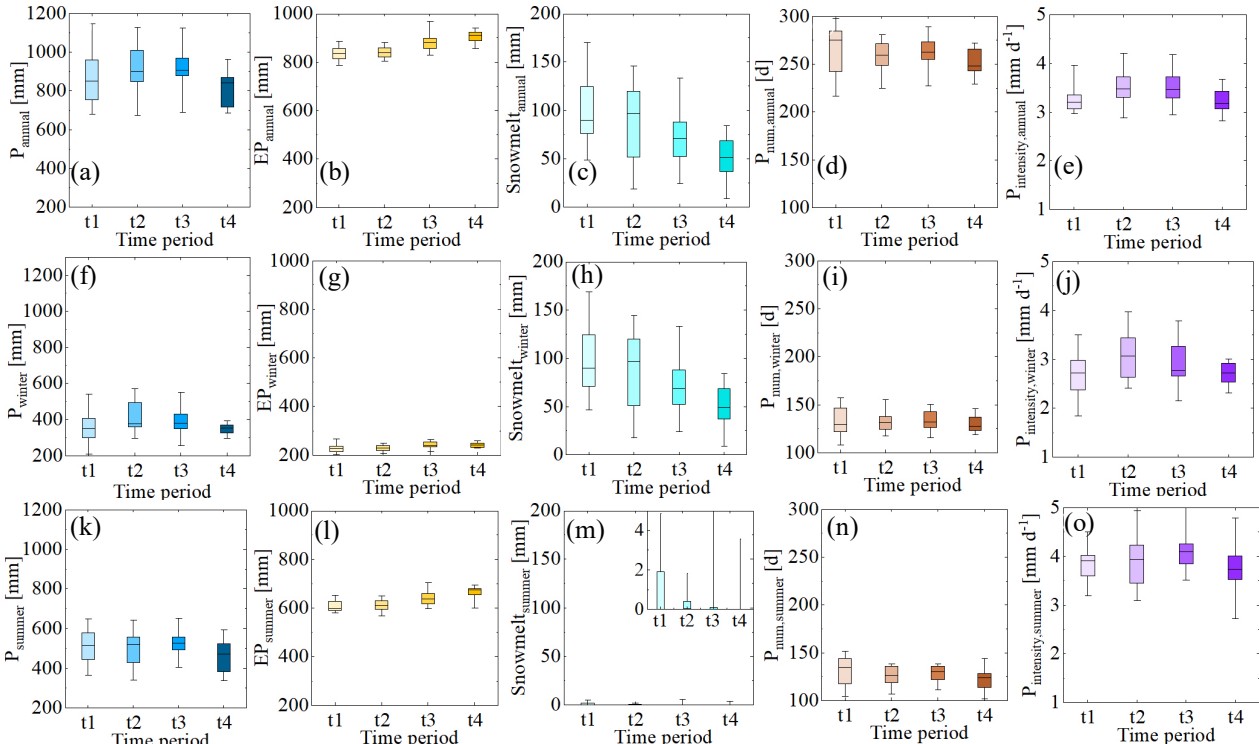

**Figure 3.** The annual and seasonal variability (i.e., winter and summer) of selected climatic indices including annual averages of precipitation (P), potential evaporation ($E_P$), estimated snow melt water, the number of precipitation days ($P_{num}$) and precipitation intensity ($P_{intensity}$) for four sub-time periods ($t_1$:1953-1972, $t_2$:1973-1992, $t_3$:1993-2012, and $t_4$:2013-2022). (a) – (e) the annual variability of selected climatic indices; (f) – (j) the seasonal variability of selected climatic indices in winter periods; (k) – (o) the seasonal variability of selected climatic indices in summer periods.

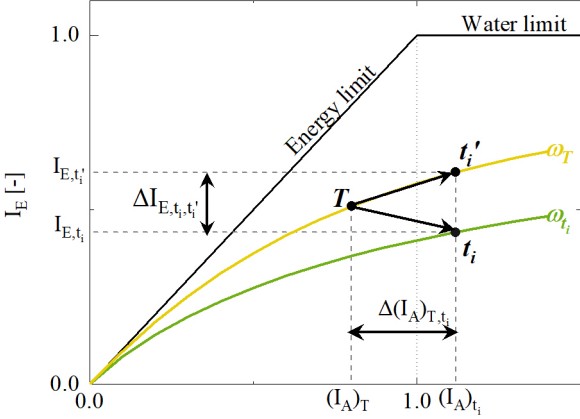

**Figure 4.** Representation of the Budyko space, which shows the evaporative index ($I_E = 1-Q/P$) as a function of the aridity index ($I_A = E_P/P$) and the water and energy limit. A catchment with the long term mean aridity index $I_{A,T} = E_{P,T}/P_T$ and evaporative index $I_{E,T} = 1-Q_T/P_T$, which is derived from observed entire-time-period data, plots at location T on the parametric Budyko curve with $\omega_T$ (yellow line) as the baseline. Based on observed sub-time-period data, with the aridity index $I_{A,ti} = E_{P,ti}/P_{ti}$ and evaporative index $I_{E,ti} = 1-Q_{ti}/P_{ti}$, the same catchment plots at location $t_i$ on the parametric Budyko curve with $\omega_{ti}$ (green line). A movement in the Budyko space towards $t_i$ 'along the $\omega_T$ curve is shown as a result of a change in the aridity index $I_{A,ti}$ with the assumption that the long-term mean Budyko curve trajectory and the parameter $\omega$ is transferable across time for an individual catchment, which results in a significant deviation $\Delta I_{E,ti,ti'}$ between the observed evaporative index $I_{E,ti}$ and the predicted evaporative index $I_{E,ti'}$.





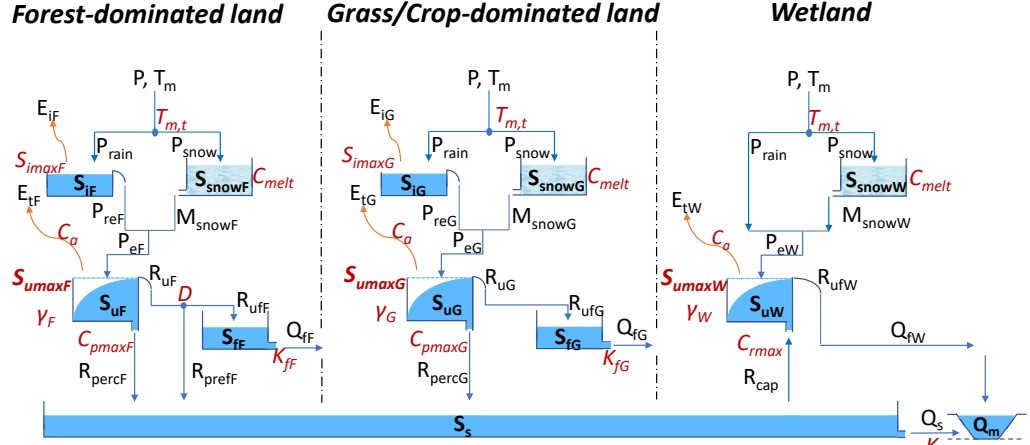

**Figure 5.** Model structure of the distributed conceptual hydrological model, discretized into three parallel hydrological response units HRU, i.e. forest, grassland and wetland in each precipitation zone P1 – P3. The light blue boxes indicate the hydrologically active individual storage volumes. The arrow lines indicate water fluxes and model parameters are shown in red. All symbols are described in Table S1 in the Supplementary Material.

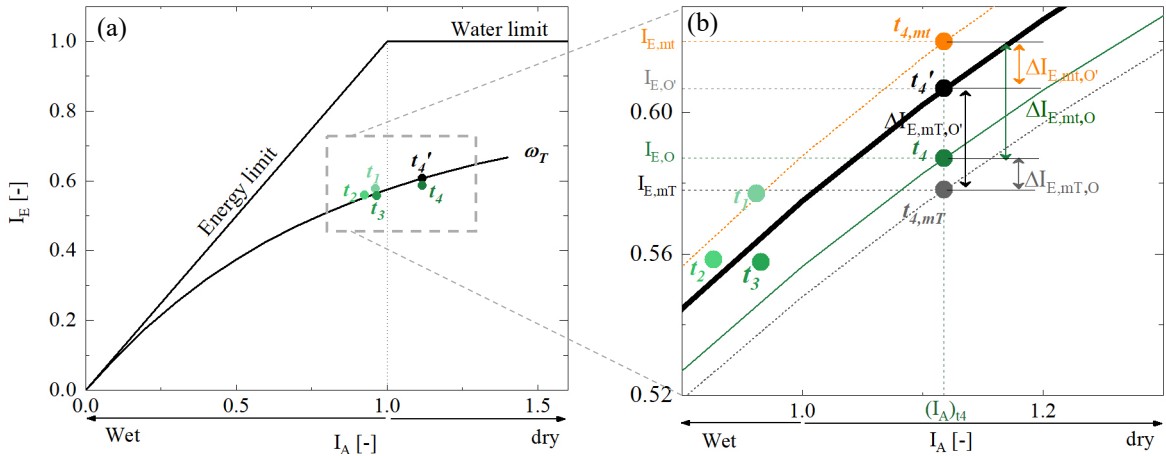

**Figure 6.** (a) Green dots from light "•" to dark "•" indicate the observed positions for four sub-time periods from $t_1$ to $t_4$. The black dot "•" $t_4'$ indicates the expected location on the parametric Budyko curve with $\omega_T$ derived from observed entire time period. We select time period $t_4$ as an example to present the modelled positions in the zoom-in plot (b). The gray dot "•" $t_{4,mT}$ indicates the modelled position based on scenario1 which is with $S_{umax,WB,T}$, and the orange dot "•" $t_{4,mt}$ indicates the modelled position based on scenario 2 which is with $S_{umax,WB,t4}$. $\Delta I_{E,mT,O'}$ (black arrow) indicates the deviation between modelled $I_{E,mT}$ ("•") based on scenario 1 and expected $I_{E,O'}$ ("•"). $\Delta I_{E,mt,O'}$ (orange arrow) indicates the deviation between modelled $I_{E,mt}$ ("•") based on scenario 2 and expected $I_{E,O'}$ ("•"). $\Delta I_{E,mT,O}$ (gray arrow) indicates the deviation between modelled $I_{E,mT}$ ("•") based on scenario 1 and observed $I_{E,O}$ ("•"). $\Delta I_{E,mt,O}$ (green arrow) indicates the deviation between modelled $I_{E,mt}$ ("•") based on scenario 2 and observed $I_{E,O}$ ("•").



**Figure 7.** (a), (c), (e), (g) and (i) The time series of storage deficits as calculated by Eq. 7, for entire time period T (1953-2022) and four sub-time periods (green shades from light to dark for time period from $t_1$ to $t_4$). The maximum annual deficits are indicated by the dots. (b), (d), (f), (h) and (j) Estimation of $S_{umax}$ as the storage deficit associated with a 40-year return period using the Gumble extreme value distribution for different time periods. The orange crosses indicate the values of $S_{umax}$ for different time periods.





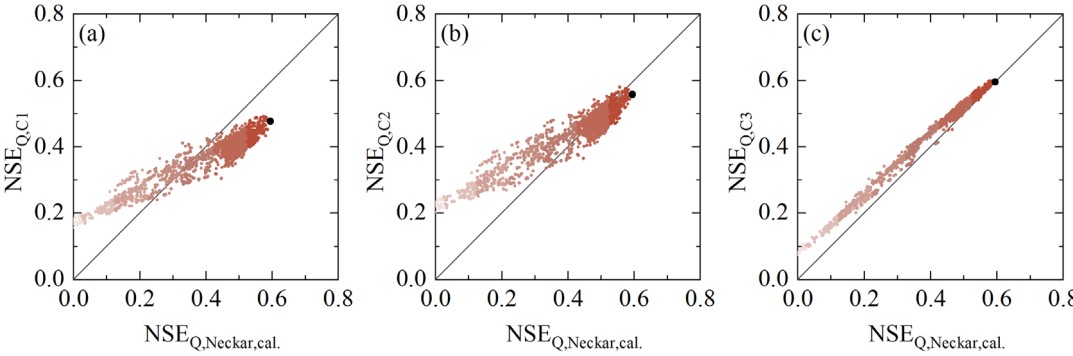

**Figure 8.** Selected model performance metrics in the entire time period 01/01/1953 – 31/12/2022 of the upper Neckar basins against the model performance in uncalibrated sub-catchment C1: Rottweil, C2: Plochingen at files river, and C3: Horb based parameter sets derived from the calibration for entire time period. The dots indicate all pareto optimal solutions in the multi-objective model performance space. The shades from dark to light indicate the overall model performance based on the Euclidean Distance $D_E$, with the black solutions representing the overall better solutions (i.e. larger $D_E$)

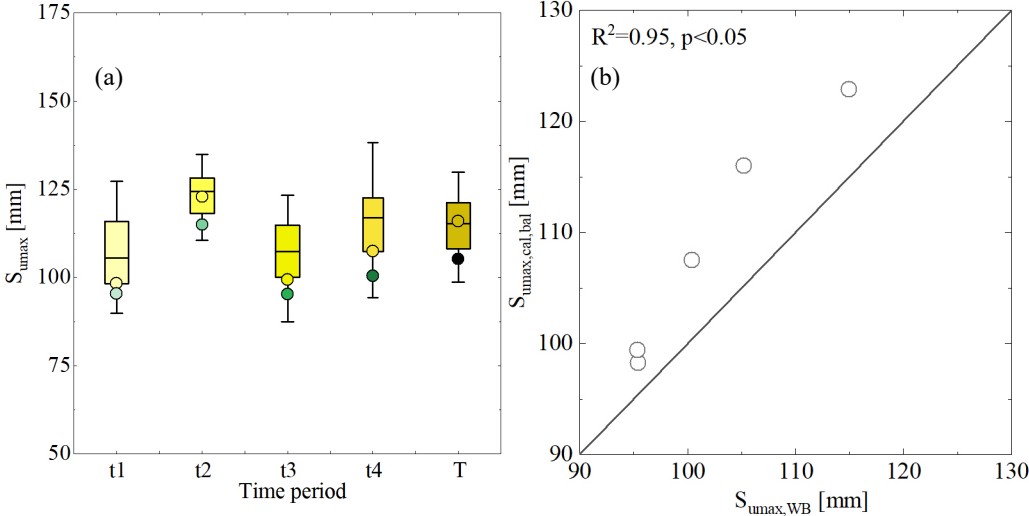

**Figure 9.** (a) $S_{umax}$ values derived from water-balance method and hydrological model for different time periods. The yellow boxes from light to dark indicate the range of $S_{umax,cal}$ for the sub-time period from $t_1$ to $t_4$ and entire time period T based on the corresponding parameter sets derived from the model, yellow dots indicate the corresponding $S_{umax,cal,bal}$ based on the most balanced solution, and green dots indicate the corresponding $S_{umax,WB}$ derived from water-balance method. (b) the values of $S_{umax,cal,bal}$ against $S_{umax,WB}$.





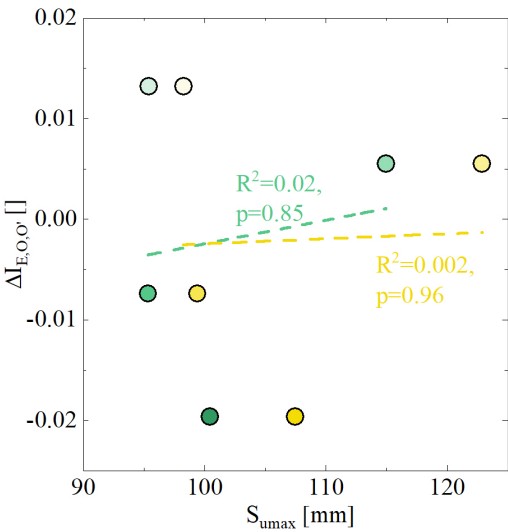

**Figure 10.** Relationships between the deviations $\Delta I_{E,O,O'}$ and the values of $S_{umax,WB}$ and $S_{umax,cal}$ for four sub-time periods ($t_1$-$t_4$) which are respectively derived from the water-balance-method (green circles) and hydrological model calibration (yellow circles).

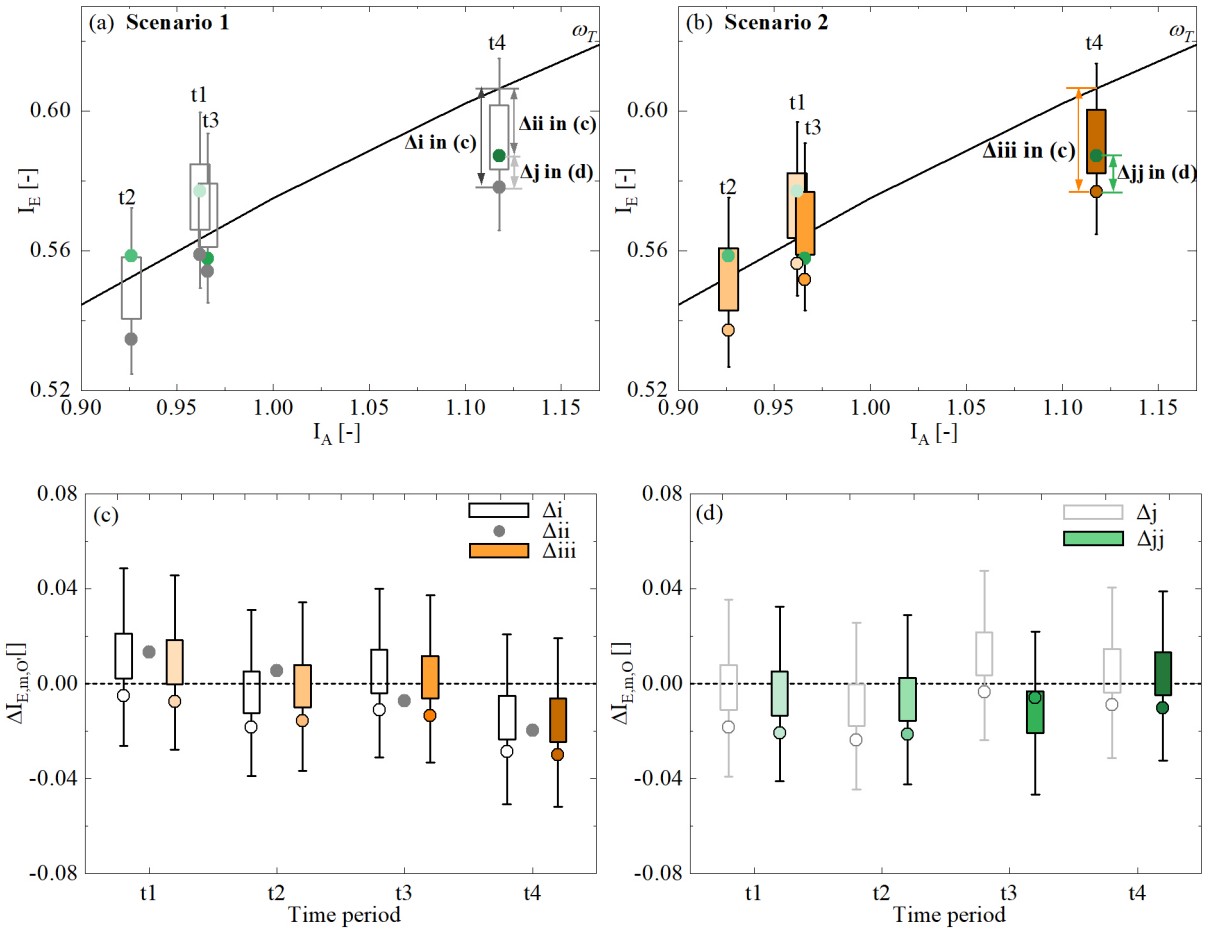





**Figure 11.** (a) The gray boxes ($I_{E,mT}$) indicate the modelled evaporative index based on all pareto front solutions for four sub-time periods based on scenario 1 with a stationary $S_{umax,WB,T}$ and gray dots based on the most balanced solution based on scenario 1. The green circles from light to dark in (a) and (b) indicate the observed evaporative index for four sub-time periods from $t_1$ to $t_4$. (b) The orange circles ($I_{E,mt}$) indicate the modelled evaporative index based on all pareto front solutions for four sub-time periods (from lighter to darker shades) based on scenario 2 with time-variant $S_{umax,WB,t_i}$ and the orange circles based on the most balanced solution for scenario 2. Black boxes in (c) indicate the deviations $\Delta I_{E,mT,O'} = I_{E,mT} - I_{E,O'}$ ($\Delta i$)based on all pareto front solutions for four sub-time periods, and the dark gray circles based on the most balanced solution based on scenario 1. Orange boxes in (c) indicate the deviations $\Delta I_{E,mt,O'} = I_{E,mt} - I_{E,O'}$ ($\Delta iii$) based on all pareto front solutions for four sub-time periods, and the orange circles based on the most balanced solution for scenario 2. The gray dots indicate the deviation $\Delta I_{E,O,O'}$ ($\Delta ii$) between observed $I_{E,O}$ for each sub-time period and corresponding expected $I_{E,O'}$. Light gray boxes in (d) indicate the deviations $\Delta I_{E,mT,O} = I_{E,mT} - I_{E,O}$ ($\Delta j$) based on all pareto front solutions for four sub-time periods, and the gray circles based on the most balanced solution. Green boxes in (d) indicate the deviations $\Delta I_{E,mt,O} = I_{E,mt} - I_{E,O}$ ($\Delta jj$) based on all pareto front solutions for four sub-time periods, and the green circles based on the most balanced solution.

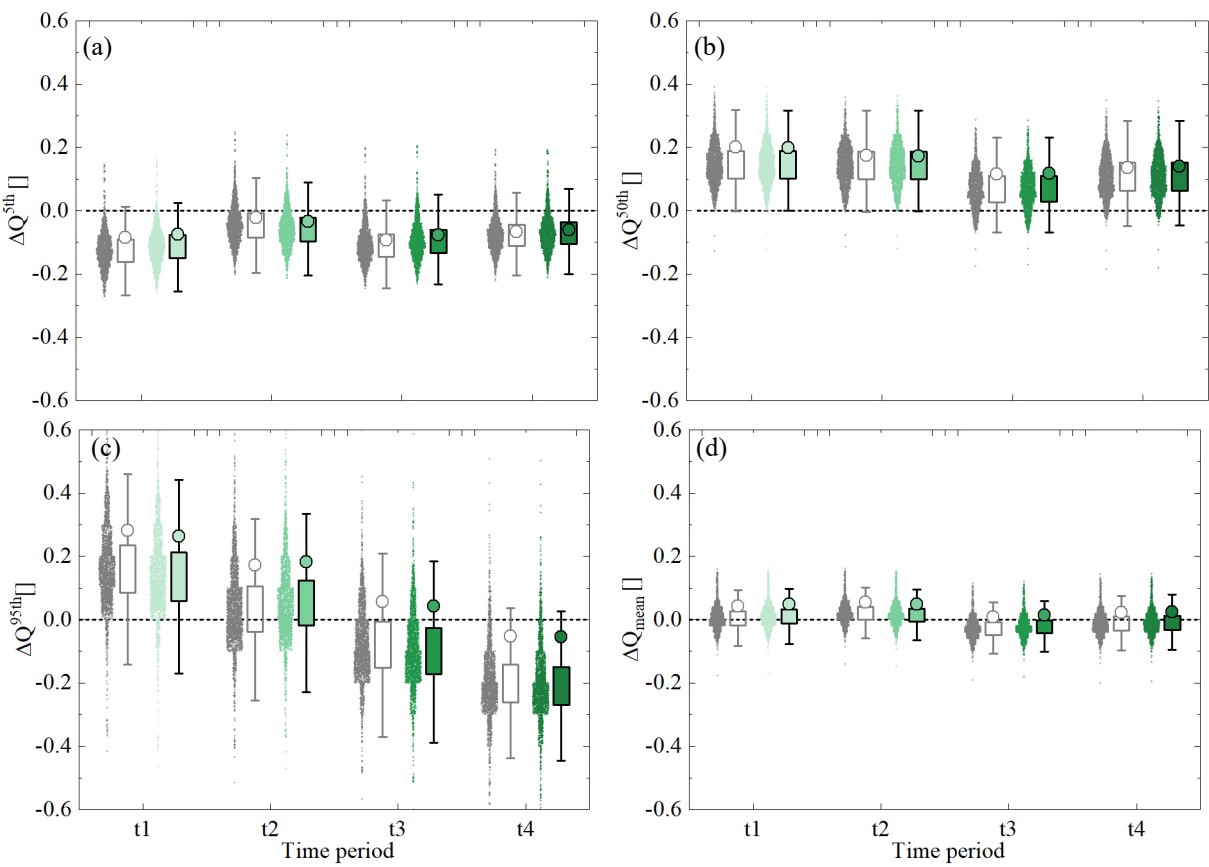

**Figure 12.** The relative errors of observed and modelled high ($Q^{5th}$), median ($Q^{50th}$), and low ($Q^{95th}$) flow quantiles and the mean Q for different time periods based on two scenarios. The gray shades and corresponding dots indicate the relative errors based on scenario 1 with all pareto front solutions and the gray circles indicate the most balanced solution. The corresponding values for scenario 2 are shown in green.



**Figure 13.** Relationships between the temporal fluctuation of the deviations ($\Delta I_{E,O,O'}$, the black dots in Fig. 10a), and $S_{umax,WB}$ and climate indices including precipitation (P), potential evaporation ($E_P$), aridity index ($I_A$), estimated snow melt water (Snowmelt), the number of precipitation days ($P_{num}$) and precipitation intensity ($P_{intensity}$) for the four sub-time periods. Relationships between the temporal fluctuation of the deviations ($\Delta I_{E,O,O'}$, the black dots in Fig. 10a) and climate indices for the four sub-time periods are shown in the first two rows (a)-(l) (i.e., (a)-(f) $\Delta I_{E,O,O'}$ vs. annual climatic indices, (g)-(l) $\Delta I_{E,O,O'}$ vs. seasonal climatic indices). Relationships between the temporal fluctuation of $S_{umax,WB}$ derived from the water-balance-method and climate indices for the four sub-time periods are shown in the last two rows (m)-(x) (i.e., (m)-(r) $S_{umax,WB}$ vs. annual climatic indices, (s)-(x) $S_{umax,WB}$ vs. seasonal climatic indices).