# Peer review of "Multi-decadal fluctuations in root zone storage capacity through vegetation adaptation to hydro-climatic variability has minor effects on the hydrological response in the Neckar basin, Germany."

_Hydrology and Earth System Sciences, 2024_

## Author Comment (AC1)

We highly appreciate the time and effort that the Reviewer has dedicated to providing feedback on our manuscript and are grateful for the insightful comments on our manuscript. Please find below our detailed replies to the individual comments.

*Major comments:*

*(1) Reviewer Comment:*

*The abstract is unnecessarily long and contains repetitive statements. I suggest making it more concise.*

**Reply:**

We completely agree with this suggestion. We will therefore reformulate concisely our abstract to make it shorter in the revised manuscript.

*(2) Reviewer Comment:*

*I am confused by the use of the terminologies "transpiration," "evaporation," and "transpiration" in certain contexts. For example, in Lines 198-205, Equation 5 should represent the overall water balance in a watershed, thus Er_bar should indicate total evapotranspiration rather than just transpiration. In Line 203, author used "potential evaporation", "evaporation". These make me lost.*

**Reply:**

Thank you for pointing this out. In Equation 5: $\overline{E_r} = \overline{P_e} - \overline{Q_O}$ , Where $\overline{P}_e$ (mm d$^{-1}$) is the long-term mean effective precipitation which is estimated by Equation 2: $P_e(t) = P(t) - E_i(t) - dS_i/dt$, where the volume of effective precipitation $P_e$ (mm d$^{-1}$) represents the fraction of the total precipitation (P) that actually reaches the soil after accounting for canopy interception and the associated interception evaporation $E_i$ (Figure FR1). Interception evaporation $E_i$ is here (Eq. 3) assumed to be equivalent to EP, as evaporation of water intercepted at the surface of leaves is effectively "open water evaporation", i.e. EP. Therefore, Er_bar in Equation 5 represents just transpiration excluding interception evaporation ($E_i$). Also note that, for brevity, although Er is referred to as transpiration throughout this paper, it also contains soil evaporation, which is comparatively minor (e.g. Brutsaert, 2014) and thus not individually accounted for here. In line 203, we clarified the estimation method of the daily transpiration (Er) based on "potential evaporation (EP)" and "interception evaporation (Figure FR1)", but we will make it clearer in the revised manuscript.

[Figure]

**Figure FR1.** The concept figure of water-balance method to estimate root zone storage capacity (Sumax).

**(3) Reviewer Comment:**

*Somehow, I am unable to access the data and model in the Code and Data Availability section. Although HESS does not mandate the openness of data/code/user-guides like the Geoscientific Model Development does, I encourage the author to make these accessible to enable readers to replicate or advance the work, thus expanding its impact.*

**Reply:**

Thank you for pointing this out. We agree, and we will upload the model code to an open access repository. All hydrological data are available via open access databases as explicitly highlighted in text and the Data availability section.

**(4) Reviewer Comment:**

*I question the solid physical meaning of the Sumax. Firstly, Sumax is not a directly measurable feature using devices; it seems to be derived from known variables (precipitation, ET, streamflow). Such derivations generally should have a clear meaning, indicating their driving factors. Hence, the question arises: what are the driving factors determining the value of Sumax? Can it be measured without long-term climate data or model calibration? The equations 1-8 calculate Srd.n rather than derive Sumax. The concept of Sumax seems more akin to a feature in a conceptual model, derived from data. Unlike field or laboratory measurable parameters like conductivity in soil flux calculations via Darcy's Law, Sumax cannot be directly measured or validated experimentally.*

**Reply:**

This is indeed a very interesting point. Sumax [mm] is here and in a wide range of previous analyses (Kleidon et al., 2004; Gentine et al., 2012; Gao et al., 2014; Wang-Erlandsson et al., 2016; Nijzink et al., 2016; Singh et al., 2020; Dralle et al., 2021; Hrachowitz et al., 2021; McCormick et al., 2021; Giardina et al., 2023; Stocker et al., 2023; Hahm et al., 2024) defined as the **maximum subsurface water volume between permanent wilting point and field capacity that is within the reach of roots and therefore accessible to plants for transpiration**. As such it is an inherently scale-independent real system property and model parameter. We fully agree that it is currently not directly observable at larger scales. However,

its controlled by the interaction between water and energy supply and the eventual adaptation of vegetation root-systems to satisfy the plant water demand that arises from that interaction. More specifically, it is the amplitudes and the phase lags between peak seasonal precipitation and solar radiation reflect this vegetation accessible subsurface water volume Sumax (e.g. Gentine et al., 2012; Stocker et al., 2023). The interaction between the amplitudes and phase lags is implicit in Eqs. 1 – 8 that underlie the estimation of Sumax (see also in the references provided above).

To put this method into a wider context, let us also consider the physical background of the root zone storage capacity Sumax. To survive dry periods, vegetation needs continuous access to water stored in the subsurface and accessible to roots to satisfy its canopy water demand. As a consequence, the vegetation present at any moment, and in particular its active root system, reflects its successful adaptation to the prevalent climatic conditions in a region (Laio et al., 2001; Schenk and Jackson, 2002; Rodriguez-Iturbe et al., 2007; Donohue et al., 2007; Gentine et al., 2012; Liancourt et al., 2012). Irrespective of geometry, distribution or structure of root systems, **Sumax [mm] and thus the maximum vegetation-accessible water storage volume in the unsaturated root zone of the subsurface**, represents the hydrologically relevant information of root systems (Rodriguez-Iturbe et al., 2007; Nijzink et al., 2016a; Savenije and Hrachowitz, 2017; Gao et al., 2024). Therefore, **the value of Sumax is directly affected by the root depth and root distributions of plants**. In response to a changing environment, the root systems of vegetation continuously adapt to allow the most efficient use of available energy and resources for surviving. The driving factors for roots adaptation are also the driving factors for Sumax, as Sumax inherently represents adaptations of the root system (i.e., the climatic factors, the soil texture, the landscape).

Based on the definition of Sumax, if our research only focuses on one plant or point study, we can estimate Sumax based on the measurements of soil porosity, root depth and root density, without long-term climate data or model calibration (see e.g. de Boer-Euser et al., 2016). However, our study focuses on a large catchment scale. It is not possible to collect the root information for each plant in our catchment, nor do we have sufficiently accurate data on soil properties. It is therefore necessary to adopt a Darwinian perspective (Harman and Troch, 2014) and to estimate effective values of Sumax to reflect the collective and adaptive behaviour of all individual plants within our catchment. Then we choose two common methods which have been used in many previous studies to estimate Sumax. From Eqs. 1 – 8 we then indeed compute the maximum water storage deficits (Srd.n). Then the estimation of the Sumax is based on the Gumble extreme distribution. Previous studies suggested that vegetation develops root zone storage capacities large enough to survive in dry spells with return periods of ~ 20 – 40 years (Gao et al., 2014; deBoer-Euser et al., 2016; Wang-Erlandsson et al., 2016; Hrachowitz et al., 2021). Therefore, we define Sumax,WB here as the maximum storage deficit in a 40-year period so that Sumax,WB=Srd,40yr.

**(5) Reviewer Comment:**

*Sumax is derived from the differences between effective precipitation and transpiration. The calibrated values of Sumax (Sumax,cal) are computed using outputs from the FLEX model, which is calibrated by streamflow. This suggests that the streamflow simulations are reliable within the FLEX model but may not imply the reliability of ET calculations (PET, AET, evaporation, transpiration, etc.). I would like to hear your thoughts on this challenge.*

**Reply:**

Thank you for pointing this out. This is a very interesting observation. Streamflow and evaporation are the most important fluxes in hydrological system. Exactly, it is necessary to make sure both of them reliable. Indeed, we calibrated our model to streamflow observations to make the modelled streamflow reliable. **Firstly, the daily potential evaporation EP** (mm d−1) was estimated using the Hargreaves equation based on the observed daily maximum and minimum temperature, which has been used in many previous studies and shown to be among the most reliable methods (Oudin et al., 2005). **Secondly, for the estimation of actual evaporation**, we used Figure 2b to check if the estimation of actual evaporation is reliable based on our model. Based on the observed long-term data of precipitation and stream flow, we calculate the long-term average actual evaporation Ea by closing the water balance Ea = P – Q, shown as evaporative index: $I_E$ = Ea/P= 1-Q/P = 0.57 (see table 2 in the manuscript) for the entire time period. The total Ea over that time period estimated from our observations is ~ 35.000 mm After calibrating our model based on the entire time period from 1953-2022, we showed the total modelled actual evaporation over 70 years is also ~35.000 mm, which means that that modelled long-term evaporation is consistent with the observations. **Finally,** we used equations 5-6 to estimate **the transpiration (Er)**, which is one part of the actual evaporation (Ea = Ei + Er), based on the long -term effective precipitation (Pe) and observed streamflow (Qo). To make the estimation of transpiration reliable, we need to make sure to have a plausible estimate of effective precipitation, which is the amount of precipitation that really reaches the soil after interception evaporation (Ei). Effective precipitation (Pe), here, is estimated by solving the water balance of an interception storage (Si) with maximum interception storage capacity (Simax; here taken as 2.0 mm). As Sumax is not very sensitive to the choice of Simax as previously shown by e.g. Hrachowitz et al., (2021) and Bouaziz et al. (2022), we used here an value of Simax = 2.0 mm, which was previously also used by de Boer-Euser et al. (2016) and Bouaziz et al. (2022).

We will clarify that in the revised manuscript.

*(6) Reviewer Comment:*

*If Sumax can be derived from observed precipitation, ET, etc., what is the necessity for model calibration? Consider a hypothetical experiment: if someone sets the Sumax value in the FLEX model based on observed data and then calibrates the model using streamflow, would this experiment yield comparable performance metrics (e.g., NSE) to those obtained from the model/simulation? I know you already test the model output via fixed Sumax, but did not focus on the NSE performances.*

**Reply:**

This is indeed an important and interesting comment. On the one hand, as you observed, the Sumax can be estimated by the water-balance method based on observed hydrological data, on the other hand, as parameter in hydrological model, Sumax can also be derived by the calibration of a hydrological model. Both of these two different methods can estimate the value of Sumax. The two methods are largely independent of each other. If therefore the values of Sumax (and their evolution over time) derived from these two methods do not conflict, but instead remain consistent, this is evidence that the estimated values of Sumax from both methods reflect at least to some degree its real-world value.

We agree that replacing the calibrated Sumax with a fixed value does result in very similar model performances without recalibration of the model. From, extensive prior model testing we found that additional calibration of the other parameters while keeping Sumax fixed does only slightly improve the model performance. This was therefore not further explored here.

In any case, we will, for completeness, add the performance metrics of the model for both cases: calibrated and fixed (i.e. water balance-derived Sumax) in Table S3 in the Supplementary Material in the revised manuscript.

**(7) Reviewer Comment:**

*An opinion paper by Gao et al. (2023) (10.5194/hess-27-2607-2023) discusses concepts that may connect to the soil features or arguments presented in this manuscript. I am neutral on the opinions expressed in Gao et al. (2023), but I am curious whether there are links between the Sumax concept and the points made in this paper.*

**Reply:**

This is a very interesting comment. Gao et al. (2023) pointed that the traditional understanding of the high importance of soil may to some extent mislead the understanding of hydrological processes and the development of hydrological model. And they suggest that we need to consider the terrestrial ecosystem structure to improve our understanding of hydrological processes and how the ecosystem can be survived and developed. Our research focuses on how the ecosystem adapts to climatic variability, reflected by the fluctuation of the Sumax values, and the effects on the long-term partitioning of drainage and evaporation and hydrological response. Therefore, there is indeed a link between Gao et al. (2023) and our research. The conclusions of Gao et al. (2023) somewhat support our research objectives and indicates that our research is meaningful. We will clarify that in the revised manuscript.

**(8) Reviewer Comment:**

*The paper attempts to establish a connection between Sumax and vegetation adaptation to climate. However, I do not see any analysis on vegetation adaptation, except for the use of omega in the Budyko method. Moreover, the ET is an output from the model, not an observation linked to vegetation-specific features. Given these uncertainties, I believe the current findings are sufficient for publication and recommend not expanding them to include vegetation adaptation.*

**Reply:**

We acknowledge that our description of the link between Sumax and vegetation adaptation in the original manuscript was not sufficiently clear. Related to the reply to comment (4), the definition of Sumax is the maximum vegetation-accessible water storage volume in the unsaturated root zone of the subsurface. Therefore, the value of Sumax is directly linked with vegetation as it defines the water volume accessible to roots and thus, by extension, the size and structure of the root system. To survive, vegetation continuously adapts to the changing environment by adapting the root depth or root density, which both

directly affect the values of Sumax. Therefore, changes in Sumax over time explicitly reflect the adaptation of vegetation root-zones to changing hydro-climatic conditions.

Concerning ET, please note that in water-balance method, we estimated ET based on observed data of precipitation and stream flow, by closing the long-term water balance. This method then requires the rationale that the surplus water resulting from long-term average P-Q needs to have been evaporated/transpired in the past. However, this can only be the case with a sufficiently large vegetation accessible storage volume Sumax – otherwise, vegetation would not have access to sufficient water in dry periods to transpire the observed water volumes.

We will further clarify that in the revised manuscript.

**(9) Reviewer Comment:**

*I encourage the author to disclose all the calibrated parameter values from the model. These values indicate both the performance of the model and the characteristics of the watersheds.*

**Reply:**

We completely agree with this suggestion. We will add all the calibrated parameter values from the model into the revised supplement.

**(10) Reviewer Comment:**

*Figure S1: The groundwater storage (Ss) in the figure implies a seasonal variation. What factors cause the seasonal variation of groundwater storage? The variation of groundwater storage implies the variation of baseflow, but it did not affect the total streamflow (Q). Is there any data/analysis support the groundwater storage and baseflow? There are two more reservoirs (unsaturated fast) in the model. Could you show the outputs about them?*

**Reply:**

Thank you for pointing this out. Here note that the groundwater storage (Ss) in our model indicates the *active* groundwater storage, not including the passive, hydrologically inactive groundwater storage, which is estimated by Wang et al. (2023), about 4000mm. The total groundwater storage including active and passive parts does not vary so much during one year. When we return to the groundwater storage (active groundwater) which is showed in Figure S1, indeed, there is a seasonal variation. The driving factors can be precipitation which is the primary input source of the groundwater recharge. In this particular case, the lower winter groundwater levels are related to the groundwater depletion during the preceding summer/autumn period and to periods of snow cover in winter, when groundwater is not (or only at low rates) being recharged. Seasonal changes in precipitation directly impact groundwater levels, with wet seasons often leading to increased groundwater recharge (Figure FR2). As you said, the slow streamflow from this component in our model structure (Figure 5 in the original manuscript) is often synonymous with baseflow, which is part of the total streamflow.

In any case, we will also show the seasonal fluctuation of storage in the unsaturated storage component in the revised supplement.

[Figure]

**Figure FR2.** The concept figure of water-balance method to estimate root zone storage capacity (Sumax).

**(11) Reviewer Comment:**

*Let's conceptualize an ideal watershed based on your presented data. When the maximum water deficit in the root zone (Sumax) is about 120 mm (your results), and the maximum groundwater storage is approximately 4 mm (Figure S1), assuming typical porosities for the two layers (p_root = 0.4 and p_gw = 0.2), the calculated depth of the average hydrological-response aquifer would be 120/0.4+4/0.2=320 mm. This value represents the aquifer depth necessary for the hydrological response in this watershed. However, this formulation does not account for the unsaturated and fast reservoirs, as these are not detailed in the manuscript. I wish to highlight two concerns: (1) the calculated 320 mm depth for the hydrological-response aquifer seems unreasonable; (2) there is a need for information and analysis concerning the unsaturated and faster reservoirs to better understand the watershed dynamics and model structure.*

**Reply:**

Thank you for pointing this out and we will try to clarify this in the following. Related to the reply to comment (10), here, the slow response storage (Ss) is only the active groundwater, not including the passive groundwater. The difference between the two is that only the active groundwater generates hydraulic heads (ha>0) and thus flow (i.e. it is the water stored in aquifers above stream water levels). Groundwater below stream water levels cannot generate hydraulic head (hp=0) and are therefore hydrologically inactive, i.e. assuming an impermeable bedrock boundary condition, they always store the same volume of water so that dS/dt = 0, as indicated in the sketch provided in FR3 below. For further details, we would like to refer you to Zuber (1986; and in particular Figure 1 therein) and Hrachowitz et al. (2016). As we estimated passive groundwater in Wang et al. (2023), the value is ~4000mm. Considering the larger hydrologically inactive, passive part of the groundwater then the estimated value of groundwater storage increases from 320mm to be ~ 20320mm which is reasonable for hydrological response.

We agree that any unsaturated zone below the root zone is not accounted for in this type of model. The reason is that with no roots present below the root zone, water cannot be extracted from these deeper layers of the subsurface by evaporation/transpiration. As a result, the soil moisture of this zone will not

go below capacity as water is held against gravity in the pores and cannot drain. After a rainfall event, the infiltrating wetting front will pass through this unsaturated zone. But as it is already at filed capacity, no additional water can be stored there and the total water volume that has entered the unsaturated zone below the root zone with the precipitation event will eventually recharge the groundwater with a few days of delay, depending on the depth of the groundwater and the soil permeability. As such, the unsaturated zone below the root zone does only temporarily store water for a few days before being released into the groundwater. It does therefore, at the much longer time-scales that regulate Sumax, not have any discernible effect. This is also the reason why standard, state-of-the-art conceptual hydrological models do not consider this additional zone (e.g. Perrin et al., 2003; Fenicia et al., 2006; Clark et al., 2008; Samaniego et al., 2010; Gharari et al., 2013; Newman et al., 2017; Seibert et al., 2022). The same applies for the consideration of passive groundwater storage as described above, which is only needed for tracer and water quality studies but not water quantity studies as in detail described by Hrachowitz et al. (2016), due to the difference in water flow velocities and celerities as described in detail by and McDonnell and Beven (2014).

[Figure]

**Figure FR3.** Sketch of the definition of hydraulically and hydrologically active (Sa) and passive (Sp) groundwater storage in a hillslope cross-section and the respective representation thereof in hydrological models (after Hrachowitz et al., 2016).

**Minor Comments**

**(12) Reviewer Comment:**

*What is the f(x) in equation 7?*

**Reply:**

Thank you for pointing this out. *f(x)* in equation 7 indicates the following equations. We will clarify this clearly in the revised manuscript.

**(13) Reviewer Comment:**

*Line205: what is the El-bar in your equation?*

**Reply:**

Thank you for pointing this out. *El-bar here indicates the long term mean interception evaporation. We will add the statement of the meaning of El-bar in the revised manuscript.*

**(14) Reviewer Comment:**

*Line219-221: citation is necessary.*

**Reply:**

We completely agree with this suggestion. We will add the relevant references in the revised manuscript.

**(15) Reviewer Comment:**

*Line790: I don't see the figure2(b) make any sense here. The accumulation of ET in such long period does not tell clear message here. The shaded areas for the t1-t4 are not very clear in these figures. I am not sure, but the maximum streamflow in figure 2d seems beyond of the y-axis-max and was crop out of the figure box.*

**Reply:**

Thank you for pointing this out. We used Figure 2b to check and demonstrate that the estimation of actual evaporation is reliable based on our model. As your comment (5) said, we can not only make sure streamflow is reliable, but also make sure the estimated actual evaporation reliable. As another important fluxes except streamflow, the actual evaporation also needs to be checked after calibrating the hydrological model. Based on the long-term water balance, we calculate the actual evaporation should be around 35.000 mm. After calibrating our model based on the entire time period from 1953-2022, we showed the total modelled actual evaporation over the same 70 years is also about 35000 mm, which means that our model worked well for estimating the actual evaporation. We will clarify this more clearly in the revised manuscript.

**(16) Reviewer Comment:**

*Table S3: I interpreted the values in the table, for example, "0.59(0.06-0.55)" in the first cell, as "mean/media (min – max)", but the value "mean" is out of range of min-max. Do I misunderstand the meanings of the value in the table?*

**Reply:**

Thank you for pointing this out. In Table S3, we showed the performance metrics for the most balanced solution and the ranges of all performance metrics for all pareto optimal solutions for two calibration cases (Scenarios 1 – 2). So, for example, *"0.59(0.06-0.55)"* indicates "the most balanced solution (5% percentile- 95% percentile)", more specifically, *"0,59"* is the value of Nash–Sutcliffe efficiency (NSE) of streamflow based on the most balanced solution *(the largest DE based on our calibration, see the specific definition of DE in 4.3.2)*. And *"0.55"* indicates the 95% percentile, not the maximum value. We will clarify this clearly in the revised supplement.

**(17) Reviewer Comment:**

*Line348: No section 4.1.2 in this manuscript.*

**Reply:**

Indeed. We will correct that.

***(18) Reviewer Comment:***

*Line 352: What is the p value here? They seem not the common p-values in statistics. I don't think the 4-sample analysis can tell any potential relationship between the two variables, let alone any convincing conclusions.*

**Reply:**

We of course completely agree with the reviewer. A 4-sample analysis is statistically not very meaningful. We do not intend to use the relationship in any quantitative way but merely intend to use it to point out the corresponding temporal evolution of the Sumax estimates from the two independent methods. However, to avoid any misunderstandings, we will remove the value from Figure 9 in the revised manuscript.

***(19) Reviewer Comment:***

*Line 471-472: "The catchment-scale root zone storage capacity (Sumax) is a critical factor affecting the moisture exchange between land and atmosphere as well as the hydrological response in terrestrial hydrological systems". The "affect" may not the right word, "reflect" may be.*

**Reply:**

Agreed. We will correct that.

***(20) Reviewer Comment:***

*Figure S1: I cannot find the gray shades in the figure. Or are they fully overlapped with green shades?*

**Reply:**

Right. Gray shades derived based on scenario 1 with time-invariant Sumax are almost fully covered by green shades derived by scenario 2 with time-variant Sumax. We will make this clearer in the revised version.

**References:**

Bouaziz, L. J., Aalbers, E. E., Weerts, A. H., Hegnauer, M., Buiteveld, H., Lammersen, R., Stam, J., Sprokkereef, E., Savenije, H. H., and Hrachowitz, M.: Ecosystem adaptation to climate change: the sensitivity of hydrological predictions to time-dynamic model parameters, Hydrology and Earth System Sciences, 26, 1295-1318, 2022.

Brutsaert, W.: Daily evaporation from drying soil: Universal parameterization with similarity, Water Resources Research, 50, 3206-3215, 2014.

Clark, M. P., Slater, A. G., Rupp, D. E., Woods, R. A., Vrugt, J. A., Gupta, H. V., Wagener, T., and Hay, L. E.: Framework for Understanding Structural Errors (FUSE): A modular framework to diagnose differences between hydrological models, Water Resources Research, 44, 2008.

de Boer-Euser, T., McMillan, H. K., Hrachowitz, M., Winsemius, H. C., and Savenije, H. H.: Influence of soil and climate on root zone storage capacity, Water Resources Research, 52, 2009-2024, 2016.

Donohue, R., Roderick, M., and McVicar, T. R.: On the importance of including vegetation dynamics in Budyko's hydrological model, Hydrology and Earth System Sciences, 11, 983-995, 2007.

Dralle, D. N., Hahm, W. J., Chadwick, K. D., McCormick, E., and Rempe, D. M.: Accounting for snow in the estimation of root zone water storage capacity from precipitation and evapotranspiration fluxes, Hydrology and Earth System Sciences, 25, 2861-2867, 2021.

Fenicia, F., Savenije, H., Matgen, P., and Pfister, L.: Is the groundwater reservoir linear? Learning from data in hydrological modelling, Hydrology and Earth System Sciences, 10, 139-150, 2006.

Gao, H., Fenicia, F., and Savenije, H. H.: HESS Opinions: Are soils overrated in hydrology?, Hydrology and Earth System Sciences, 27, 2607-2620, 2023.

Gao, H., Hrachowitz, M., Schymanski, S., Fenicia, F., Sriwongsitanon, N., and Savenije, H.: Climate controls how ecosystems size the root zone storage capacity at catchment scale, Geophysical Research Letters, 41, 7916-7923, 2014.

Gao, H., Hrachowitz, M., Wang-Erlandsson, L., Fenicia, F., Xi, Q., Xia, J., Shao, W., Sun, G., and Savenije, H. H.: Root zone in the Earth system, EGUsphere [preprint], https://doi.org/10.5194/egusphere-2024-332, 2024.

Gentine, P., D'Odorico, P., Lintner, B. R., Sivandran, G., and Salvucci, G.: Interdependence of climate, soil, and vegetation as constrained by the Budyko curve, Geophysical Research Letters, 39, 2012.

Gharari, S., Hrachowitz, M., Fenicia, F., Gao, H., and Savenije, H.: Using expert knowledge to increase realism in environmental system models can dramatically reduce the need for calibration, Hydrology and Earth System Sciences, 18, 4839-4859, 2014.

Giardina, F., Gentine, P., Konings, A. G., Seneviratne, S. I., and Stocker, B. D.: Diagnosing evapotranspiration responses to water deficit across biomes using deep learning, New Phytologist, 240, 968-983, 2023.

Hahm, W., Dralle, D., Lapides, D., Ehlert, R., and Rempe, D.: Geologic controls on apparent root-zone storage capacity, Water Resources Research, 60, e2023WR035362, 2024.

Harman, C. and Troch, P.: What makes Darwinian hydrology" Darwinian"? Asking a different kind of question about landscapes, Hydrology and Earth System Sciences, 18, 417-433, 2014.

Hrachowitz, M., Stockinger, M., Coenders-Gerrits, M., van der Ent, R., Bogena, H., Lücke, A., and Stumpp, C.: Reduction of vegetation-accessible water storage capacity after deforestation affects catchment travel time distributions and increases young water fractions in a headwater catchment, Hydrology and earth system sciences, 25, 4887-4915, 2021.

Hrachowitz, M., Benettin, P., Van Breukelen, B. M., Fovet, O., Howden, N. J., Ruiz, L., Van Der Velde, Y., and Wade, A. J.: Transit times—The link between hydrology and water quality at the catchment scale, Wiley Interdisciplinary Reviews: Water, 3, 629-657, 2016.

Kleidon, A. and Lorenz, R. D.: Non-equilibrium thermodynamics and the production of entropy: life, earth, and beyond, Springer Science & Business Media2004.

Laio, F., Porporato, A., Ridolfi, L., and Rodriguez-Iturbe, I.: Plants in water-controlled ecosystems: active role in hydrologic processes and response to water stress: II. Probabilistic soil moisture dynamics, Advances in water resources, 24, 707-723, 2001.

Liancourt, P., Sharkhuu, A., Ariuntsetseg, L., Boldgiv, B., Helliker, B. R., Plante, A. F., Petraitis, P. S., and Casper, B. B.: Temporal and spatial variation in how vegetation alters the soil moisture response to climate manipulation, Plant and Soil, 351, 249-261, 2012.

McCormick, E. L., Dralle, D. N., Hahm, W. J., Tune, A. K., Schmidt, L. M., Chadwick, K. D., and Rempe, D. M.: Widespread woody plant use of water stored in bedrock, Nature, 597, 225-229, 2021.

McDonnell, J. J. and Beven, K.: Debates—The future of hydrological sciences: A (common) path forward? A call to action aimed at understanding velocities, celerities and residence time distributions of the headwater hydrograph, Water Resources Research, 50, 5342-5350, 2014.

Newman, A. J., Mizukami, N., Clark, M. P., Wood, A. W., Nijssen, B., and Nearing, G.: Benchmarking of a physically based hydrologic model, Journal of Hydrometeorology, 18, 2215-2225, 2017.

Nijzink, R., Hutton, C., Pechlivanidis, I., Capell, R., Arheimer, B., Freer, J., Han, D., Wagener, T., McGuire, K., and Savenije, H.: The evolution of root-zone moisture capacities after deforestation: a step towards hydrological predictions under change?, Hydrology and Earth System Sciences, 20, 4775-4799, 2016.

Oudin, L., Hervieu, F., Michel, C., Perrin, C., Andréassian, V., Anctil, F., and Loumagne, C.: Which potential evapotranspiration input for a lumped rainfall–runoff model?: Part 2—Towards a simple and efficient potential evapotranspiration model for rainfall–runoff modelling, Journal of hydrology, 303, 290-306, 2005.

Perrin, C., Michel, C., and Andréassian, V.: Improvement of a parsimonious model for streamflow simulation, Journal of hydrology, 279, 275-289, 2003.

Rodríguez-Iturbe, I. and Porporato, A.: Ecohydrology of water-controlled ecosystems: soil moisture and plant dynamics, Cambridge University Press2007.

Samaniego, L., Kumar, R., and Attinger, S.: Multiscale parameter regionalization of a grid-based hydrologic model at the mesoscale, Water resources research, 46, 2010.

Savenije, H. H. and Hrachowitz, M.: HESS Opinions Catchments as meta-organisms–a new blueprint for hydrological modelling, Hydrology and Earth System Sciences, 21, 1107-1116, 2017.

Schenk, H. J. and Jackson, R. B.: Rooting depths, lateral root spreads and below-ground/above-ground allometries of plants in water-limited ecosystems, Journal of Ecology, 480-494, 2002.

Seibert, J. and Bergström, S.: A retrospective on hydrological catchment modelling based on half a century with the HBV model, Hydrology and Earth System Sciences, 26, 1371-1388, 2022.

Singh, C., Wang-Erlandsson, L., Fetzer, I., Rockström, J., and Van Der Ent, R.: Rootzone storage capacity reveals drought coping strategies along rainforest-savanna transitions, Environmental Research Letters, 15, 124021, 2020.

Stocker, B. D., Tumber-Dávila, S. J., Konings, A. G., Anderson, M. C., Hain, C., and Jackson, R. B.: Global patterns of water storage in the rooting zones of vegetation, Nature geoscience, 16, 250-256, 2023.

Wang-Erlandsson, L., Bastiaanssen, W. G., Gao, H., Jägermeyr, J., Senay, G. B., Van Dijk, A. I., Guerschman, J. P., Keys, P. W., Gordon, L. J., and Savenije, H. H.: Global root zone storage capacity from satellite-based evaporation, Hydrology and Earth System Sciences, 20, 1459-1481, 2016.

Wang, S., Hrachowitz, M., Schoups, G., and Stumpp, C.: Stable water isotopes and tritium tracers tell the same tale: no evidence for underestimation of catchment transit times inferred by stable isotopes in StorAge Selection (SAS)-function models, Hydrology and Earth System Sciences, 27, 3083-3114, 2023.

Zuber, A.: On the interpretation of tracer data in variable flow systems, Journal of Hydrology, 86, 45-57, 1986.

---

## Author Comment (AC2)

We highly appreciate this positive overall assessment of our work and we thank the reviewer for her or his interest in our work as well as for the thoughtful comments that helped to strengthen our analysis. Below, we provide clarifications and our perspectives to respond in detail to the individual reviewer comments.

*(1) Reviewer Comment:*

*The authors divided the whole period into four subperiod to calculate the Sumax, its relation with climatic indices, and its influence on hydrological response. However, I would question whether such a division could produce reasonable results. First, many climatic indices don't show significant difference among four periods, making it difficult to see the relation between these indices and the Sumax. Second, regression based on only four points has large uncertainty and occasionality. For example, in Figure 10, if we remove the point with largest Sumax, a significant negative relation between ΔIE and Sumax can be obtained. Maybe the authors can attempt to increase the number of subperiod or discuss this issue in a limitation section.*

**Reply:**

Thank you for pointing this out. We indeed divided the entire period into four sub-periods. The maximum vegetation-accessible water storage volume in the unsaturated root zone of the subsurface is the definition of root zone storage capacity Sumax (see first paragraph in the original manuscript). To be survived, the root systems of vegetation and the associated vegetation-accessible water storage capacity (Sumax) are therefore at a dynamic equilibrium with and responding to the ever-changing conditions of its environment. However, as these changes occur at landscape scale and are mostly reflected by the composition of plant species present in a specific spatial domain, the changes occur at time-scales that reflect the life-cycles of individual plants. Thus, periods of at least 20-years are required to reflect this and for meaningful estimates of Sumax, as also demonstrated by many other studies (e.g. Gao et al., 2014; Lan et al., 2016; Singh et al., 2020; Stocker et al., 2023). We therefore had to strike a balance between the number of independent time periods and the robustness of the associated Sumax estimates. We deliberately chose to emphasize fewer but longer time periods and thus rather reliable estimates of Sumax.

However, we positively acknowledge and agree with the point raised by the reviewer. We will add some discussion of this limitation in the revised version of the manuscript.

*(2) Reviewer Comment:*

*For most of figures, I cannot see the necessity of using gradual color to distinct the results of different period, since they can be clearly distanced by the x-axis. Instead, for Figure 9b, I think showing the period of each point by different color would be better.*

**Reply:**

Thank you for pointing this out. We agree that it is not necessary to use gradient colour scheme for some figures, as already clear enough based on their different values. However, we still prefer to make the readers more clearly aware of the difference between each dot when they just see the figures. We completely agree with your suggestion about Figure 9b. We will change that colour scheme in the revised manuscript.

**(3) Reviewer Comment:**

*There are lots of variables in this paper. I would like to suggest the authors to provide a table to show the meanings of all the variables to make the paper easier to follow. Besides, if I don't miss something, I think some variables are not explained. (1) Equation 7 is confusing. What does f(x) mean? What does Srd(t) and what is the difference from Srd,n(t)? The meaning of n is not explained. (2) The subscripts o and o' described in 5.4 haven't appeared in the method section. I guess it may be explained in the missing 4.1.2 section.*

**Reply:**

Thank you for pointing this out, this is indeed an excellent suggestion. The *f(x)* in equation 7 indicates a symbol of general function and it is equal to the following equations. There should not be *Srd(t),* and we missed *n* in *Srd(t)* here in equation7.And *n* here indicates one specific year. And it is our fault to make you confused about 4.1.2 section. We will correct this and clarify all the variables clearly and consistently in a table in the revised manuscript.

**(4) Reviewer Comment:**

*As pointed out by another reviewer, the abstract is too long. The three paragraphs are actually telling one thing, that is, the three hypotheses and the related to them. I also suggest the authors to change the expression of the hypotheses to the form of scientific question, at least for the first paragraph of abstract. I was really confused when I read the second hypothesis for the first time because it was contradictory to the title, and finally I realize that it is just a hypothesis which is rejected later. I think express them more straightly could help readers get your main conclusions more easily.*

**Reply:**

We completely agree with this suggestion. We will reformulate concisely our abstract to make it shorter and clear in the revised manuscript.

**(5) Reviewer Comment:**

*For the Sumax determined by hydrological model, the authors regarded all parameters on the pareto front as feasible. However, there are some extremely low values for some metrics such as NSEQ and NSElogQ. I think it would be better to select the behavioral solutions based on the threshold of each metric for analysis. Also, I would like to suggest the authors to present the metrics for each subperiod produced by scenario 1,*

*and that for the whole period T produced by scenario 2 in Table S3, to allow for a direct comparison between two scenarios.*

**Reply:**

Thank you for pointing this out. Although we presented all feasible pareto front solutions to show the uncertainty of our model, we already chose the most balanced solution based on the overall performance metric described by the Euclidian distance (DE) (see 4.3.1 section). In any case, the choice of which solutions to keep as feasible will always have to have a subjective aspect. In particular, for sets of Pareto optimal solutions there are multiple ways to deal with that as in detail described by e.g. Efstratiadis and Koutsoyiannis (2010) or Gharari et al. (2013). We deliberately chose to use all solutions on the Pareto front to obtain a conservative estimate of uncertainty.

We will clarify that in the revised manuscript and we will add the performance metrics for each time period based on scenario 1 in the revised supplement.

*(6) Reviewer Comment:*

*Although the calculation and analysis are solid, the main conclusion of this paper is not so favorable for its publication. The results indicate that the change of Sumax neither controls the drainage/evaporate water flux partitioning, nor affects short term hydrological response dynamics, and considering the variation of Sumax also leads to little improvement in hydrological model performance. So a reader may question why we need to care about Sumax. I suggest the authors to add some open discussion on the significance of Sumax and its influence on hydrological cycle. Besides, given that the conclusion is different to some other studies, it is strongly recommended to discuss what factors determine whether the hypotheses 2 and 3 would be rejected, i.e., in what kind of catchments, considering the change of Sumax would improve model performance? This will make the conclusion of this paper more general and useful.*

**Reply:**

This is an interesting comment. Indeed, the results in our paper imply that the temporal evolution of Sumax does not control variation in the partitioning of water fluxes and has no significant effects on fundamental hydrological response characteristics of the Upper Neckar basin during time period from 1953-2022 (see Conclusion section in the original manuscript). As the statements in our conclusion, we already said this conclusion is limited in our study basin, in a cool-temperature climate with ample summer precipitation. This combination does not only lead to rather low Sumax, but also implies that in such an environment, where sufficient precipitation is available during the periods of highest canopy water demand (i.e. highest EP, and thus summer) Sumax is of minor relevance: The much less pronounced effects on hydrological response we found in our analysis are a consequence of the rather low absolute magnitude of Sumax that remains below 115 mm in the study region. These low Sumax values reflect lower storage requirements in summer, due to a precipitation pattern in the Neckar basin that is more evenly spread throughout the year. In other words, the fact that here ~55 − 60 % of the annual precipitation falls in summer (Fig. 3f, k in the original manuscript) when it is needed most by vegetation due to high EP, removes the need for larger Sumax as water storage buffer to allow vegetation to survive. However, the lower the magnitude of Sumax, the more frequently storage deficits can be overcome by even rather small rainstorms and the less water is (or needs to be) stored. Even if the relative changes are

similar between Bouaziz et al. (2022) in a somewhat more humid catchment and our study, abundant summer precipitation causes absolute Sumax fluctuations of less than ±20 mm over time in the Neckar. This in turn limits the influence of the changes on the hydrological response, which has wider implications on the use of models in the Neckar basin and potentially in other temperate regions with similar hydro-climatic characteristics (see 6.2 section). This in itself is already an interesting finding as it gives modellers process-based evidence that the use of time-invariant Sumax as model parameter will be also sufficient for meaningful predictions over at least the next few decades in such environment. However, it also needs to be expected that in more arid regions with less summer precipitation, where Sumax is higher (see e.g. Gao et al., 2014; Stocker et al., 2023) changes in Sumax will play a much more prominent role.

 We totally agree that a more detailed discussion of which reasons cause the less pronounced effects in our study and potentially more pronounced effects in other environments will be helpful for the reader. We will thus expand on the discussion in the revised version.

**References:**

Bouaziz, L. J., Aalbers, E. E., Weerts, A. H., Hegnauer, M., Buiteveld, H., Lammersen, R., Stam, J., Sprokkereef, E., Savenije, H. H., and Hrachowitz, M.: Ecosystem adaptation to climate change: the sensitivity of hydrological predictions to time-dynamic model parameters, Hydrology and Earth System Sciences, 26, 1295-1318, 2022.

Efstratiadis, A. and Koutsoyiannis, D.: One decade of multi-objective calibration approaches in hydrological modelling: a review, Hydrological Sciences Journal–Journal Des Sciences Hydrologiques, 55, 58-78, 2010.

Gao, H., Hrachowitz, M., Schymanski, S., Fenicia, F., Sriwongsitanon, N., and Savenije, H.: Climate controls how ecosystems size the root zone storage capacity at catchment scale, Geophysical Research Letters, 41, 7916-7923, 2014.

Gharari, S., Hrachowitz, M., Fenicia, F., and Savenije, H.: An approach to identify time consistent model parameters: sub-period calibration, Hydrology and Earth System Sciences, 17, 149-161, 2013.

Singh, C., Wang-Erlandsson, L., Fetzer, I., Rockström, J., and Van Der Ent, R.: Rootzone storage capacity reveals drought coping strategies along rainforest-savanna transitions, Environmental Research Letters, 15, 124021, 2020.

Stocker, B. D., Tumber-Dávila, S. J., Konings, A. G., Anderson, M. C., Hain, C., and Jackson, R. B.: Global patterns of water storage in the rooting zones of vegetation, Nature geoscience, 16, 250-256, 2023.

Wang-Erlandsson, L., Bastiaanssen, W. G., Gao, H., Jägermeyr, J., Senay, G. B., Van Dijk, A. I., Guerschman, J. P., Keys, P. W., Gordon, L. J., and Savenije, H. H.: Global root zone storage capacity from satellite-based evaporation, Hydrology and Earth System Sciences, 20, 1459-1481, 2016.